# UNIVERSAL CONCAVITY-AWARE DESCENT RATE FOR OPTIMIZERS

## ABSTRACT

Many machine learning problems involve a challenging task of calibrating parameters in a computational model to fit the training data; this task is especially challenging for non-convex problems. Many optimization algorithms have been proposed to assist in calibrating these parameters, each with its respective advantages in different scenarios, but it is often difficult to determine the scenarios for which an algorithm is best suited. To contend with this challenge, much work has been done on proving the rate at which these optimizers converge to their final solution, however the wide variety of such convergence rate bounds, each with their own different assumptions, convergence metrics, tightnesses, and parameters (which may or may not be known to the practitioner) make comparing these convergence rates difficult. To help with this problem, we present a minmax-optimal algorithm and, by comparison to it, give a single descent bound which is applicable to a very wide family of optimizers, tasks, and data (including all of the most prevalent ones), which also puts special emphasis on being tight even in parameter subspaces in which the cost function is concave.

## 1 INTRODUCTION

Many machine learning problems involve calibrating the parameters of a given model to match the data distribution of a phenomenon one wishes to model, e.g. the structure of folded proteins, processing images to automatically generate appropriate labels for them, or generating images and text to interactively chat with a human engagingly. This process involves:

1. Collecting many samples ("data points") from the desired data distribution.

2. Measuring how well the model fits the collected data points (the "data set") with a given performance analysis metrics (the "loss function", a.k.a. the "objective function"). By convention, lower values of the loss function imply better performance on the model's part.

3. Adjusting the model's parameters to improve the performance, as measured by the loss function ("model parameter optimization").

4. Repeat until desired performance achieved.

However, no single existing optimizer is best suited to all machine learning problems - each has its unique strengths and weaknesses (see Vaswani et al. (2020); Sivan et al. (2024); Ruder (2016); Mustapha et al. (2021); Bera & Shrivastava (2020); Zeiler (2012); Duchi et al. (2011b); Xu et al. (2017); Wadia et al. (2021); Mittal et al. (2019); Zhou et al. (2020); Schmidt et al. (2021)), such as generalization capability, convergence rate, saddle-point and flat region evasion capability, robustness to hyperparameter choice, computational complexity per-iteration, memory complexity, etc., and different areas in which it empirically seems to work best. As a result, one must compare among various different optimization algorithms (henceforth, "optimizers") to select the one most suited to the current scenario.

In an effort to help practitioners select the best optimizer for their setup and estimate the absolute computational resources that will be required to obtain a given performance, many experiments have been run comparing the performance of different optimizers on a variety of applications (Xu et al., 2017; Schmidt et al., 2021), and on the theoretical side - convergence rate bounds have been proven for various different optimizers. However, due to the wide variety of assumptions, convergence rate

metrics, bound parameters (which may be expensive - if not impossible - to compute ahead of time), and tightness of the bounds in all of these works, comparing among them remains a challenging task. Secondly, there is a lack of convergence rate bounds general enough to be easily applicable to newly proposed optimizers. Thirdly, many of these bounds fail to demonstrate the empirically-verified convergence rate superiority of the more sophisticated methods that make use of second-order curvature information instead of exclusively the gradient. Lastly, although convergence rate bounds exist for non-convex functions, many of them fail to properly address the opportunities that lay in linear subspaces of the parameter space in which the loss function is concave (meaning that a restriction $f|_{\mathbb{S}} : \mathbb{S} \to \mathbb{R}$ of the loss function $f$ to a linear subspace $\mathbb{S}$ is locally concave). We believe that more attention should be given to these subspaces of the function in the context of neural network optimization; Alain et al. (2018) and Ghorbani et al. (2019) demonstrate experimentally that there is much to be gained by taking optimal steps in these subspaces, often even orders of magnitude greater than the potential gains in convex subspaces.

**Our contributions**    In an effort to help practitioners select the best optimizer for their use case, we develop a tool for estimating the value of second-order optimization algorithms; this will help decide if the additional computational burden of these algorithms is worthwhile. We develop a minmax-optimal algorithm, rate algorithms by similarity to it, and demonstrate in theory and in practice that in general, second-order algorithms work best on mechanistically simple problems. Our algorithm-optimality bound satisfies the following good properties:

1. **Concave tightness** Our bound exploits the opportunity for greater descent in subspaces of the parameter space in which the loss function is concave.

2. **Universality** We make only weak and commonly satisfied assumptions for our bound, to allow for its application to a wide and prevalent family of optimizers and loss functions.

3. **Tightness for any level of iteration step-quality** instead of assuming a bound on the quality of steps given in each iteration as some previous works have done, our theoretical bounds are given as a continuous function of the quality of each iteration's step.

4. **Bound on loss function descent** Our main result bounds the rate at which the model's performance increases (as measured by the loss function). This is in contrast to previous works, which instead bound various indicators of local minimality, such as gradient norm, local near-convexity, or proximity to a local minimum (in Euclidean distance). Although Xu et al. (2020) write that the latter convergence rate metrics is more relevant to the non-convex optimization setting, we feel that the former is more practically useful, since generally real-world applications with limited computational resources simply demand a minimal performance guarantee of their model, without regard to the theoretical capabilities of a given model or optimization algorithm.

5. **Simplicity of cubic minimization problem** We approach the multidimensional cubic polynomial minimization problem posed by Nesterov & Polyak (2006) by decomposing it into $n$ 1-dimensional problems via eigendecomposition of the Hessian, making our approach to the solution of this minimization problem far simpler conceptually.

Our paper is organized as follows: In section 2, we review previous work and describe the notation we will use throughout the paper. In section 3, we develop the minmax-optimal `ELMO` algorithm and analyze its descent rate. In section 4, we make claims as to the benefits of optimizer similarity to `ELMO` (proven in appendix H). In section 5 we show the value of our novel Lipschitz parameter separation scheme by showing how much lower the Lipschitz parameters of most relevant eigenspaces can be, thus giving optimizers a more accurate minimizable model of the loss function in each neighborhood it finds itself in. Finally, in section 6, we present experiments validating one particular use case of our bound: we show that the advantage second-order optimizers hold over first-order optimizers is inversely proportional to the convex Lipschitz parameters. In other words, second-order optimizers present strong performance (thus may be worth their additional computational burden) in settings with small convex Lipschitz parameters, and weak performance (thus not worthwhile) in settings with large convex Lipschitz parameters.

## 2 BACKGROUND

**Assumption 1.** *For a given optimization problem with loss function $f : \mathbb{R}^n \to \mathbb{R}$, we assume $f$ is twice differentiable.*

We note that this assumption is satisfied for all prevalent deep learning optimization problems for all but a zero-measure set of parameters.

### 2.1 NOTATIONS AND DEFINITIONS

**Notation 1.** Let $\theta_{t+1}, \theta_t \in \mathbb{R}^n$ the parameter vectors of a pair of consecutive iterations of a given optimization algorithm.

- For brevity of notation, we mark $\theta_{t+1} - \theta_t \triangleq \Delta\theta_t$.

- We mark $\nabla f(\theta_t)$ the gradient of $f$ and $\mathcal{H}(\theta_t)$ the Hessian of $f$ at $\theta_t$.

**Notation 2.** Let $\theta_t \in \mathbb{R}^n$. We mark $(v_i(\theta_t), \lambda_i(\theta_t))_{j=0}^n$ an orthogonal eigendecomposition of $\mathcal{H}(\theta_t)$ (which exists due to the Hessian symmetry property). For brevity of notation, we will sometimes drop the $(\theta_t)$ and just write $v_i, \lambda_i$ when the meaning is clear.

Since $v_i$ and $-v_i$ are both equally viable eigenvectors, we eliminate ambiguity by assuming

$$\forall_{i \in [n]} : \nabla f(\theta_t)^\top v_i \leq 0 \tag{1}$$

**Definition 1.** We say an algorithm is a *k-order algorithm* if it requires oracle access to the first $k$ derivatives of $f$.

**Notation 3.** Let $A, B \in \mathbb{R}^{n \times n}$. We use the following notations (when applicable):

- We mark $A$'s transpose as $A^\top$.

- We write $A \succeq 0$ iff $A$ is positive semi-definite, $A \succ 0$ if $A$ is positive definite, $A \succeq B$ if $A - B \succeq 0$ (and likewise for $A \succ B$).

- Mark $\lambda_{\min}(A), \lambda_{\max}(A)$ the minimal/maximal eigenvalue of $A$, respectively, and their ratio $\kappa(A) = \frac{\lambda_{\max}(A)}{\lambda_{\min}(A)}$ the condition number of $A$.

**Notation 4.** For $\tau \in \mathbb{N}$ we mark $[\tau] = \{t \in \mathbb{N} : t \leq \tau\}$.

**Definition 2.** Let $U, D \in \mathbb{R}^{n \times n}$ s.t. $D = diag(d_1, d_2, \ldots, d_n)$ is diagonal and $U$ orthogonal, and let $\xi : \mathbb{R} \to \mathbb{R}$. We mark $\xi(U \cdot D \cdot U^\top) = U \cdot diag(\xi(d_1), \xi(d_2), \ldots, \xi(d_n)) \cdot U^\top$.

**Definition 3.** We say that an optimization algorithm is a *Quasi-Newton optimization algorithm* if its characteristic update rule may be expressed as:

$$\theta_{t+1} = \theta_t - \alpha_t \Phi_t \nabla f(\theta_t)$$

for $\Phi_t \in \mathbb{R}^{n \times n}, \Phi_t \succeq 0, \Phi_t^\top = \Phi_t, \alpha_t \in \mathbb{R}^+$. We call $\Phi_t$ in such algorithms the "preconditioner matrix".

This approach is inspired by Newton's method in convex optimization (see Nocedal & Wright (2006, Chapter 3)) where $\Phi_t = (\mathcal{H}(\theta_t))^{-1}$. See appendix A for a discussion of the challenges and proposed solutions involved in these algorithms.

We note that the overwhelming majority of gradient-based optimizers may be expressed as quasi-Newton optimizers (some popular examples may be seen in Martens (2020)). As a result, this paper will concern itself exclusively with this family of optimizers.

**Notation 5.** Throughout this paper, we will mark the point a convergent quasi-Newton algorithm converges to by $\theta^*$.

### 2.2 RELATED WORK

As discussed in item 4 of the contributions section, the value of the loss function after $t$ iterations is of particular importance to practitioners, due to its implications on the quality of model. One measure

of optimizer quality relating to this value is the objective function sub-optimality gap (OFSOG), defined as $f(\theta_T) - f(\theta^*)$. The ARC algorithm is a second-order algorithm that uses a low-rank SVD approximation of the Hessian and estimates a single Hessian-Lipschitz parameter adaptively; Cartis et al. (2012b) prove that OFSOG-optimality (bounding the OFSOG to below $\epsilon$) is achieved by a variant of the ARC algorithm after $\mathcal{O}\left(\epsilon^{-1}\right)$ iterations in the convex regime, or $\mathcal{O}\left(\log\left(\epsilon^{-1}\right)\right)$ iterations in the strongly convex regime. Garmanjani (2020) show similar bounds for the Nonlinear Stepsize Control algorithm family, and Toint (2013) demonstrate that this is a generalization of ARC and trust-region methods. Liu et al. (2024) prove OFSOG-optimality for the Sophia optimizer (a second-order algorithm that approximates the Hessian as a diagonal matrix, which is estimated with Hutchinson's estimator (Hutchinson, 1989)) after $\mathcal{O}\left(\epsilon^{-1}\right)$ iterations in the convex regime.

Bottou (2004) split the process of optimization with a general optimizer into the initial "search phase", in which the optimizer searches for an approximately convex region in which the point it will eventually converge to resides, and the later "final phase", in which the optimizer converges to its final solution within this convex region.

In the machine learning literature, many common loss functions are "empirical risk functions" - that is, loss functions which can be written as a sum of terms, each of which is a function of only a single sample from the data distribution. When this sum ranges over a very large number of samples, a common approach to estimating it is to perform a Monte Carlo approximation, summing over only a small subset of the terms; this approach is known as the "minibatch approach". Amari (1998) then note that when using this approach, $\theta_t$ may be seen as a statistical estimator for $\theta^*$. Working in the "final phase" (and thus assuming convexity), and adopting the estimator approach to $\theta_t$ taken by Amari (1998); Bottou & Lecun (2004) give a convergence rate bound for this estimator's variance parameterized by the first- and second-order derivatives at $\theta^*$, assuming only that $\lim_{t\to\infty} \Phi_t = \mathcal{H}^{-1}(\theta^*)$. Martens (2020) takes these convergence rates and plugs them into a Taylor approximation of $f(\theta_t)$ to obtain the asymptotic OFSOG, given by $f(\theta_T) - f(\theta^*) = \frac{n}{2T} + o\left(\frac{1}{T}\right)$.

Since the goal of optimization is to minimize a loss function, arguably the best metric for measuring an optimization algorithm's quality are the gains it makes as measured by the loss function values, i.e. its rate of loss function descent. Nevertheless, most algorithms' convergence rate bounds relate to their gradient norms; we note, however, that a bound on an algorithm's gradient norm may be a poor proxy for its descent rate in the early, nonconvex "search" phase, since convergence rate bounds may only imply proximity to a critical point of the gradient, which is neither guaranteed to be the point the algorithm will ultimately converge to nor even to have a small loss function value by any measure. To the best of our knowledge, our bound is the first to directly address the problem of bounding the loss function value in the "search" phase without assuming convexity (which is rarely satisfied by the loss functions in neural network optimization scenarios).

We refer the reader to appendix B for discussion on previous attempts at universal convergence rate bounds, other convergence rate measures, and the effect of the preconditioner on convergence rate.

## 3 A MINMAX HESSIAN LIPSCHITZ-AWARE OPTIMIZATION ALGORITHM

Any deterministic optimization algorithm is comprised of two parts: first, we gather information about the loss function to enable us to implicitly construct a local model of the loss function, and secondly we step to the minimum of this model. Accordingly, gradient descent and Newton's method use first- and second-order Taylor approximations of $f$ respectively, and while these models do give a direction of descent in every subspace of the domain space, they do not indicate optimal step sizes in concave subspaces of the domain space (that is, subspaces in which the loss function is concave), since concave first- and second-order polynomials have no minima. To obtain a unique step in all settings (so that our optimizer will be sufficiently general to apply to nonconvex and nonquadratic regions of neural network loss functions), we must therefore model $f$ with a third-order Taylor polynomial.

### 3.1 GENERAL BOUNDS ON PER-ITERATION DESCENT

A recurring theme in the neural network optimization literature is that the greatest-magnitude eigenvalues of the Hessian are slow to change, as well as their eigenvectors; see, for instance, Sivan et al. (2024); Alain et al. (2018); Sagun et al. (2016); Ghorbani et al. (2019); Gur-Ari et al. (2018);

Liu et al. (2024). It is common to formalize this as an assumption (see, e.g., O'Leary-Roseberry et al. (2019); Nesterov & Polyak (2006)) of Hessian-Lipschitz continuity with the matrix spectral norm:

$$\exists_{L_H \in \mathbb{R}} \forall_{\theta, \varphi \in \mathbb{R}^n} : \|\mathcal{H}(\theta) - \mathcal{H}(\varphi)\|_2 \leq L_H \cdot \|\theta - \varphi\|_2 \tag{2}$$

This assumption relies on a single scalar $L_H \in \mathbb{R}$ to describe the the entire Hessian's rate of change. With $\frac{n^2}{2}$ independent entries, however, the Hessian can shift in a far more subtle manner, leading this assumption to be overly conservative, requiring a very large $L_H$ for the assumption to be satisfied, leading to looseness in convergence rate bounds and subpar performance of algorithms that rely on this scalar. We instead make the following finer-grained assumption on the rate of change of the Hessian's eigendecomposition:

**Assumption 2.** *Hessian Lipschitz-Continuity in each Eigenspace*

*For any $\theta, \varphi \in \mathbb{R}^n$, let (eigendecompositions) $\mathcal{H}(\theta) = V \cdot \Lambda \cdot V^\top, \mathcal{H}(\varphi) = \tilde{V} \cdot \tilde{\Lambda} \cdot \tilde{V}^\top$ with $V, \tilde{V} \in \mathbb{R}^{n \times n}$ orthogonal matrices and $\Lambda = diag(\lambda_i)_{i=1}^n, \tilde{\Lambda} = diag\left(\tilde{\lambda}_i\right)_{i=1}^n \in \mathbb{R}^{n \times n}$ diagonal matrices, sorted s.t. $\forall_{i \in [n-1]} : \lambda_i \leq \lambda_{i+1}, \tilde{\lambda}_i \leq \tilde{\lambda}_{i+1}$. Then the following are satisfied:*

$$\forall_{\theta \in \mathbb{R}^n} \exists_{(\bar{L}^i)_{i=1}^n \in (\mathbb{R}^+)^n} \forall_{\varphi \in \mathbb{R}^n} : \left| \lambda_i - \tilde{\lambda}_i \right| \leq \bar{L}^i \cdot \left| (\theta - \varphi)^\top v_i \right|$$

$$\forall_{\theta \in \mathbb{R}^n} \exists_{L_R \in \mathbb{R}^+} \forall_{\varphi \in \mathbb{R}^n} : \left\| V - \tilde{V} \right\|_2 \leq L_R \cdot \|\theta - \varphi\|_2$$

$$\exists_{L_H \in \mathbb{R}} \forall_{\theta \in \mathbb{R}^n} \forall_{i \in [n]} : \max\left\{ L_R, \bar{L}^i \right\} \leq L_H \wedge \bar{L}^i \geq L_H^{-1}$$

When $\theta$ is the $t$-th iterate $\theta_t$ of an optimization algorithm, we'll mark the corresponding Lipschitz parameters as $L_t^i$. We will experimentally demonstrate the value of this finer assumption later, by demonstrating that these parameters vary widely. In particular, and taking into account that optimization primarily occurs in a very limited subspace of the domain space (Gur-Ari et al., 2018), we will demonstrate that the Lipschitz parameters relevant to these subspaces are often orders of magnitude smaller than the others.

The above assumption allows us to bound the loss function in each eigenspace of the Hessian; these bounds will then be applicable as tight (since the bounds satisfy assumptions 1 and 2) pessimistic and optimistic models of the loss function in the neighborhood of some iterate $\theta_t$:

**Notation 6.** Let $\theta_t \in \mathbb{R}^n$, $v_i \in \mathbb{R}^n$ an eigenvector of $\mathcal{H}(\theta_t)$.

$$M_t^i(x) \triangleq \nabla f(\theta_t)^\top v_i \cdot x + \frac{v_i^\top \mathcal{H}(\theta_t) v_i}{2} \cdot x^2 + \frac{L_t^i}{6} \cdot |x|^3 \tag{3}$$

$$m_t^i(x) \triangleq \nabla f(\theta_t)^\top v_i \cdot x + \frac{v_i^\top \mathcal{H}(\theta_t) v_i}{2} \cdot x^2 - \frac{L_t^i}{6} \cdot |x|^3 \tag{4}$$

**Lemma 3.1.** *Eigenspace Descent Bounds*

*Let $f : \mathbb{R}^n \to \mathbb{R}$ be a function satisfying assumptions 1 and 2, and let $\theta_{t+1} \in \mathbb{R}^n$. Marking $\Delta\theta_t = \theta_{t+1} - \theta_t$, we have*

$$\exists_{(L_t^i)_{i=1}^n \in (\mathbb{R}^+)^n} : f(\theta_{t+1}) - f(\theta_t) \leq \sum_{i=1}^n M_t^i\left(\Delta\theta_t^\top v_i\right) \tag{5}$$

$$\exists_{(L_t^i)_{i=1}^n \in (\mathbb{R}^+)^n} : f(\theta_{t+1}) - f(\theta_t) \geq \sum_{i=1}^n m_t^i\left(\Delta\theta_t^\top v_i\right) \tag{6}$$

## 3.2 EXPLOITING THESE BOUNDS FOR A MINMAX ALGORITHM

To gain perspective on the upcoming algorithm as a minmax algorithm, we restate a special case of the above lemma as follows: $M_t^i$ is the pointwise maximal function satisfying assumptions 1 and 2:

$$M_t^i(x) = \max_{\substack{\tilde{f}: \mathbb{R} \to \mathbb{R} \\ \tilde{f}(\theta_t) = f(\theta_t)}} \tilde{f}(\theta_t + x \cdot v_i(\theta_t))$$

Since each element of the sum is a 1-dimensional trinomial, the minmax step is now easily obtained (due to orthogonality of the eigenspaces) by taking the positive root of each term's derivative:

$$\Delta\theta_t^{*\top} v_i \triangleq \arg\min_{\Delta\theta_t} \sum_{i=1}^n M_t^i \left(\Delta\theta_t^\top v_i\right) = \frac{\sqrt{\lambda_i^2 + 2L_t^i \left|\nabla f\left(\theta_t\right)^\top v_i\right|} - \lambda_i}{L_t^i} \tag{7}$$

Finally, we are ready to present algorithm Eigenspace-Lipschitz Minmax Optimizer (ELMO). We mark `EIGEN` an eigendecomposition subroutine and `LIPSCHITZ` a Lipschitz parameter oracle.

An important observation to make about the algorithm above is its equal applicability to convex and concave regions of the domain space. In fact, when $\lambda_i < 0$ (implying a concave subspace), the step size (and, correspondingly, the amount of descent on our model of the loss function $M_t^i$) is actually greater than otherwise. This is due to ELMO's ability to make use of concave regions of the loss function for greater descent.

---

**Algorithm 1** Algorithm `ELMO`

---

**Require:** $\epsilon \in \mathbb{R}^+, \theta_0 \in \mathbb{R}^n, \texttt{EIGEN}, \texttt{LIPSCHITZ}$
  $t \leftarrow 0$
  **while** $f\left(\theta_t\right) - f\left(\theta^*\right) > \epsilon$ **do**
    $(\lambda_i, v_i)_{i=1}^n \leftarrow \texttt{EIGEN}\left(\mathcal{H}\left(\theta_t\right)\right)$
    $\left(L_t^i\right)_{i=1}^n \leftarrow \texttt{LIPSCHITZ}\left(\theta_t, (v_i)_{i=1}^n\right)$
    $\left(\Delta\theta_t^i\right)_{i=1}^n \leftarrow \frac{\sqrt{\lambda_i^2 + 2L_t^i |\nabla f(\theta_t)^\top v_i|} - \lambda_i}{L_t^i}$
    $\theta_{t+1} \leftarrow \sum_{i=1}^n \Delta\theta_t^i \cdot v_i$
    $t \leftarrow t + 1$
  **end while**

---

### 3.3 Algorithm ELMO's descent rate

An important factor in deciding how much computational power to put into optimizing a model is the ratio between the cost of computational resources and the improvement to the model's quality. To that end, we demonstrate that an upper bound on algorithm ELMO's performance has quickly diminishing rewards for additional iterations. Counter-intuitively, this is a good thing - it means that as long as the algorithm converges to an acceptable minimum point, just a few iterations are likely to be necessary in practice - since any more than that will not have much of an effect on the model's quality anyway.

**Theorem 3.2.** *Worst case-optimal descent rate Let $f$ be a function with Lipschitz-continuous Hessian. After $t$ iterations, algorithm* ELMO *satisfies*

$$f\left(\theta_0\right) - f\left(\theta_t\right) = \mathcal{O}\left(\log t\right) \tag{8}$$

Although the above theorem gives only an upper bound on the model's performance, we demonstrate that it is actually within a constant multiplicative factor of the algorithm's lower bound.

**Theorem 3.3.** *Let $f : \mathbb{R}^n \to \mathbb{R}$ satisfying assumptions 1 and 2. Algorithm* ELMO *satisfies*

$$\left|m_t^i\left(\Delta\theta_t^{*\top} v_i\right)\right| \le 5\left|M_t^i\left(\Delta\theta_t^{*\top} v_i\right)\right|$$

## 4 Descent rate of quasi-Newton optimization algorithms

Although algorithm ELMO is optimal among first- and second-order methods in the sense that its model of the loss function is a generalization of Quasi-Newton methods' and Gradient Descent's models (since its leading coefficient is not assumed to be nonzero) and its model minimization step is unique, its greater computational burden of computing the Lipschitz parameters may cause it to be an ineffective optimization algorithm in practice. Since most prevalent practical optimizers today belong to the Quasi-Newton family, we satisfy ourselves with a quantification of their quality based on their similarity to this ideal algorithm.

### The minmax preconditioner

Since Quasi-Newton methods are characterized by their preconditioners, we must first develop algorithm ELMO's characteristic preconditioner. We begin by defining a metric of distance between optimization algorithms by the difference between their characteristic steps, and find the preconditioner matrix whose corresponding quasi-Newton algorithm is equivalent to algorithm ELMO.

**Notation 7.** For a given algorithm with step $\Delta\theta_t$ at iteration $t$, mark $\Delta\Delta^i\theta_t = \Delta\theta_t^\top v_i - \Delta\theta_t^{*\top} v_i$ the step's distance from `ELMO`'s step. Since $\Delta\Delta^i\theta_t$ is a function of the algorithm chosen, it is a function of that algorithm's defining preconditioner: $\Delta\Delta^i\theta_t = \Delta\Delta^i\theta_t(\Phi_t)$

**Lemma 4.1.** *Minmax preconditioner*

*Let $f : \mathbb{R}^n \to \mathbb{R}$ satisfying assumptions 1 and 2. The preconditioner of the quasi-Newton algorithm that is equivalent to* `ELMO` *(meaning $\left|\Delta\Delta^i\theta_t\right| = 0$) is*

$$
\underset{\Phi_t \in \mathbb{R}^{n\times n}}{\arg\min} \left|\Delta\Delta^i\theta_t\left(\Phi_t\right)\right| = \left( \frac{\mathcal{H}\left(\theta_t\right) + \sqrt{\left(\mathcal{H}\left(\theta_t\right)\right)^2 + 2V \cdot diag\left(L_t^i \cdot \left|\nabla f\left(\theta_t\right)^\top v_i\right|\right)_{i=1}^n \cdot V^\top}}{2} \right)^{-1}
$$

This preconditioner shows the mechanistic similarity of our algorithm to Newton's method: while Newton's method's preconditioner is simply the inverse Hessian (which may not be positive definite), the matrix whose inverse is our algorithm's preconditioner is an average between the Hessian and a positive definite, regularized version of the Hessian, whose every eigenvalue is no less than the corresponding Hessian eigenvalue's magnitude. This ensures positive semi-definiteness of our preconditioner, with regularization dependent on the loss function's rate of curvature shift.

In fact, Newton's algorithm may even lead to a worst-case *decrease* in model quality, even when the associated loss function is convex, for sufficiently great curvature shift (measured by Lipschitz parameter). Plugging Newton's step into equation 3 and rearranging tells us that $\forall_{i\in[n]s.t.\lambda_i\geq 0}$ :

$M_t^i \left( \frac{\left|\nabla f(\theta_t)_t v_i\right|}{\lambda_i} \right) \geq 0$ for any step $t$ and eigenspace $i$ with $L_t^i \geq -3\frac{\lambda_i^2}{\left|\nabla f(\theta_t)^\top v_i\right|}$

### 4.1 PER-ITERATION DESCENT OF ARBITRARY STEP

Due to the computational difficulty of computing `ELMO`'s iteration step precisely, practitioners may prefer computationally cheaper alternatives. To address this, we provide guarantees for the worst-case rate of loss function descent of an arbitrary optimization algorithm relative to algorithm `ELMO`'s descent, as a function of the algorithm's similarity to `ELMO`. For simplicity, we restrict our discussion to the descent of the loss function's restriction to a given eigenspace $span\left(v_i\right)$.

**Notation 8.** Mark $\Delta\Delta^i\theta_t' = \frac{\Delta\Delta^i\theta_t}{\Delta\theta_t^{*\top} v_i}$ the step's distance from `ELMO`'s step relative to `ELMO`'s step.

**Theorem 4.2.** *Worst-case descent rate for arbitrary optimizers*

*Let $f : \mathbb{R}^n \to \mathbb{R}$ a twice-differentiable function satisfying assumptions 1 and 2, and let $\Delta\theta_t$ satisfy $M_t^i\left(\Delta\theta_t^\top v_i\right) \leq 0$. Then*

$$
\left| \frac{M_t^i\left(\Delta\theta_t^\top v_i\right)}{M_t^i\left(\Delta\theta_t^{*\top} v_i\right)} \right| = \Theta\left(1 + \left|\Delta\Delta^i\theta_t'\right|^2\right)
$$

$$
\left| \frac{m_t^i\left(\Delta\theta_t^\top v_i\right)}{m_t^i\left(\Delta\theta_t^{*\top} v_i\right)} \right| = \Theta\left(1 + \left|\Delta\Delta^i\theta_t'\right|^p\right) \tag{9}
$$

*with* $p = \begin{cases} 2 & \lambda_i > 0 \wedge \frac{\left|\nabla f(\theta_t)^\top v_i\right|}{\lambda_i^2} = 0 \\ 1 & else \end{cases}$.

### 4.2 GENERALIZATION OF PREVIOUS QUASI-NEWTON PRECONDITIONER QUALITY METRICS

**Notation 9.** Taking $(\lambda_i)_{i=1}^n$ the eigenvalues of $\mathcal{H}\left(\theta\right)$ for some $\theta$, note that since $n$ is finite, there exist $L^+ \triangleq \max_i\{L^i : \lambda_i > 0\}, L^- \triangleq \max_i\{L^i : \lambda_i \leq 0\}$.

Since most prevalent quasi-Newton algorithms apply a principled approach only to the concave subspaces of the loss function domain space and when the curvature shift is negligible, we examine

the special case of our metric when $\lambda_i > 0$ (when the loss function is concave over the domain subspace under examination) and show that our quality metric for quasi-Newton algorithm steps generalizes previous metrics. When $\lambda_i > 0$, we have

$$\left| \Delta\Delta^i \theta'_t \right| = \left| 1 - \frac{\nabla f\left(\theta_t\right)^\top}{\nabla f\left(\theta_t\right)^\top v_i} \cdot \left(\alpha_t \Phi_t \mathcal{H}\left(\theta_t\right)\right) \cdot v_i \cdot \frac{\sqrt{1 + 2L_t^i \cdot \frac{\left|\nabla f(\theta_t)^\top v_i\right|}{\lambda_i^2}} + 1}{2} \right| \qquad (10)$$

Županski (1993) introduce the "Effective Hessian" (a.k.a. the "Preconditioned Hessian") as $\mathcal{I}_t = \alpha_t \Phi_t \mathcal{H}\left(\theta_t\right)$, with its condition number used as a quality metric for preconditioners; ideally, $\kappa\left(\mathcal{I}_t\right) < \kappa\left(\mathcal{H}\left(\theta_t\right)\right)$. The Effective Hessian may be plainly seen in equation 10.

Mark $r_t \triangleq \left(I - \mathcal{H}\left(\theta_t\right) \cdot \Phi_t\right) \cdot \frac{\nabla f\left(\theta_t\right)}{\nabla f\left(\theta_t\right)^\top v_i}$; this is the 1-dimensional version of the quality metric $\eta_t$ for $\Phi_t$ used by Nocedal & Wright (2006, Chapter 7.1) and mentioned in appendix B (now redefined by projecting $\nabla f\left(\theta_t\right)$ onto the $i$-th eigenspace instead of taking its full norm). When $L^+ \approx 0$ (i.e. when the loss function curvature shift is negligible), equation 10 simplifies to

$$\left| \Delta\Delta^i \theta'_t \right| \approx \left| r_t^\top \cdot v_i \right|$$

## 5 LIPSCHITZ DISTRIBUTION

Previous works using the Hessian Lipschitz continuity assumption (e.g. ARC (Nesterov & Polyak, 2006) and its variants, O'Leary-Roseberry et al. (2019)) assume a single Lipschitz parameter for all eigenspaces. Although a finite number $n$ of eigenspaces ensures that such a Lipschitz parameter exists (the maximal Lipschitz parameter), they fail to account for the distribution of these Lipschitz parameters over the eigenspaces. We claim that these parameters vary widely both over the eigenspaces and over the course of training, so that a single constant value fails to capture this structure; in this section we provide evidence for this claim.

One source of interest in this distribution is for optimization algorithms (e.g. ARC) that make use of these parameters for the loss function modelling stage of each iteration. This may reduce computational complexity by reducing the number of parameters one must compute at each iteration, however appendix D shows that poorly estimating the Lipschitz parameters can have a detrimental effect on an algorithm's descent rate (thereby increasing the number of iterations the algorithm will require to converge).

Another source of interest in these parameters' distribution is in explaining the effectiveness of second-order quasi-Newton algorithms that implicitly assume the Lipschitz parameters are insignificant (i.e. very close to zero), since their model of the loss function is a quadratic Taylor polynomial (i.e. no curvature shift); this may be seen from equation 10 which shows optimality of Newton's method only when $\lambda_i > 0$ and $L^+ = 0$. We will show that they are not generally small by any means, however we will show that the Lipschitz parameters of the subspaces in which they work (the convex subspaces - see the implementation of Sivan et al. (2024), for instance, which applies Newton's method only on subspaces with significantly convex subspaces) are in fact small in certain settings.

### 5.1 EXPERIMENTS

The first source of evidence for our claim is from existing literature on the subject; we defer a discussion of this to appendix E. To test our claim directly, we modify an ARC implementation (Simpson & Wang, 2023) to compute the steps called for by `ELMO` at each point reached by a quasi-Newton algorithm, restricted to the subspace spanned by the eigenvectors corresponding to the single most positive and single most negative eigenvalues of each Hessian, and to use distinct Lipschitz parameters for each. Due to the computational difficulty of computing Lipschitz parameters precisely, we use these Lipschitz parameter values as an estimate for $L^+, L^-$. We note the crudeness of these adaptive measurements, merely adapting to keep $\frac{f(\theta_t) - f(\theta_{t+1})}{\left|\sum_{i=1}^n M_t^i\right|}$ within a given range with a restriction to powers of 2; nevertheless, the point is made.

A detailing of our experiment settings is given in appendix F as well as the full set of our experiment results, however we present two experiments in figure 1 for completeness. Our experiments show

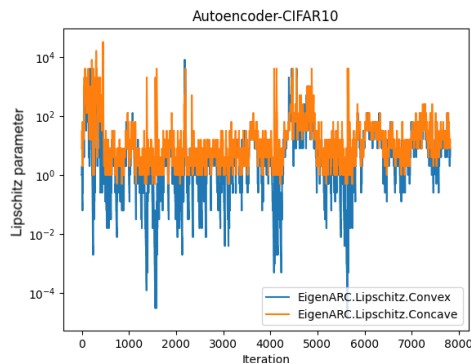 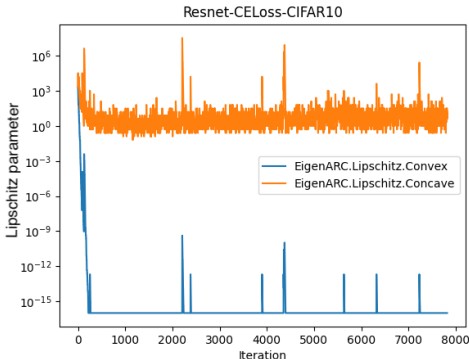

Figure 1: Comparisons of convex-subspace Lipschitz parameters to concave-subspace Lipschitz parameters. *Logarithmic scale*

that as expected, $L^+ \ll L^-$, and the gap widens exponentially as training progresses in all cases except the autoencoders. Since we will see that small convex Lipschitz parameters imply effective second-order optimization, this justifies common practice as noted by, e.g. O'Leary-Roseberry et al. (2019), of requiring the preconditioner to be an increasingly better approximation of the inverse Hessian (by increasing the strictness of the inverse Hessian approximation algorithm's stopping condition) as training progresses. Interestingly, the Lipschitz parameters seem to depend primarily on the task, and are much less affected by network structure or model output-target loss function.

Several factors seem to impact the size (by orders of magnitude) of the convex Lipschitz parameters, and they seem to be correlated with an intuitive sense of the difficulty of the setting being trained.

- The convex Lipschitz parameters are many orders of magnitude greater in the autoencoder task than in the classification task. We ascribe this gap to the more difficult task of learning a generative representation of the data instead of merely a discriminative representation of it (see Ng (2012, Chapter 4) for a discussion on generative vs. discriminative models).

- The convex Lipschitz parameters are reduced approximately 100x in the image classification task by adding residual connections. It is well known that residual connections reduce training difficulty (Li et al., 2018).

- The convex Lipschitz parameters are approximately 100x smaller when training ResNet to perform classification of natural images instead of Gaussian noise with random labels. We ascribe this to greater difficulty involved in discriminating noise, which requires partial memorization of the training set.

## 6 A QUALITY PREDICTOR FOR NEWTON'S METHOD

Expanding on the latter application in section 5, an important challenge is finding the best balance between per-iteration computational burden and expected loss function descent. We set out to provide such a metric due to equation 10 by showing that the expected descent in a given eigenspace is an approximately monotonically decreasing function of the corresponding Lipschitz parameter.

Figure 2 shows an example of this phenomenon by plotting the quasi-Newton superiority (how much better a quasi-Newton method will work than a first-order method, defined as $\left(f\left(\theta_t\right) - f\left(\theta_{t+1}^{Newton}\right)\right) - \left(f\left(\theta_t\right) - f\left(\theta_{t+1}^{SGD}\right)\right)$) against the convex Lipschitz parameter rank. Here too we represent the full spectrum of convex Lipschitz parameters with the single Lipschitz parameter representing the eigenspace with the greatest eigenvalue; nevertheless, a qualitative inverse correlation is clear. Pearson correlation coefficient values (Pearson, 1895) are shown in table 1, as well as p-values of a test of the null hypothesis that the distributions underlying the samples are uncorrelated and normally distributed. The Scipy manual writes:

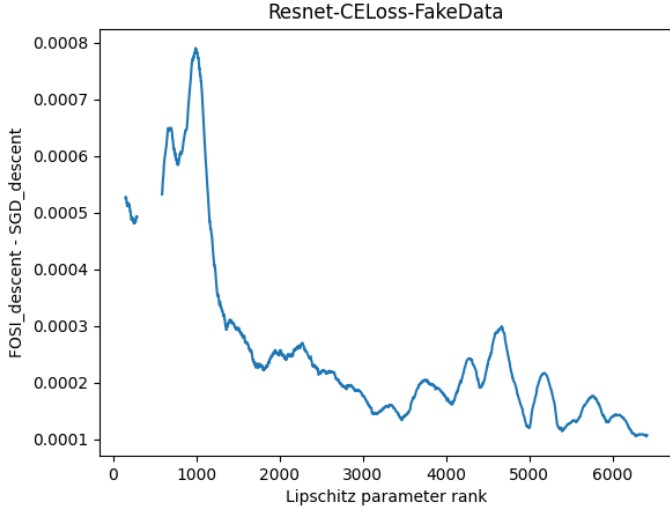

Figure 2: Inverse relation between a convex-subspace Lipschitz parameter and corresponding descent superiority of Quasi-Newton method

> The p-value roughly indicates the probability of an uncorrelated system producing datasets that have a Pearson correlation at least as extreme as the one computed from these datasets.

Here too, the detailing of our experiment settings is given in appendix F, as well as further detailing on figure 2.

Since the Lipschitz parameters are approximately locally constant throughout training as shown in the previous section, this reverse correlation may be used to help practitioners decide how much computational burden is worth putting into each iteration, given that even an exact Newton step may not be significantly superior to first-order methods when the curvature drift (as measured by Lipschitz parameters) is significantly large; hyperparameter optimization

| Dataset | Pearson $r$ | $p$-value |
|---------|-------------|-----------|
| CIFAR10 | -0.245341 | $10^{-107}$ |
| FakeData | -0.026608 | 0.031120 |
| MNIST | -0.368788 | $10^{-300}$ |

Table 1: Pearson $r$ inverse correlation between quasi-Newton superiority and Lipschitz parameter

algorithm selection may then follow accordingly. We present experiments validating this selection method in appendix G. Alternatively, practitioners may choose to use ARC steps instead of first-order methods, when the Lipschitz parameter is significantly large. These findings may instead be used to construct a meta-optimizer, that periodically computes Lipschitz parameters and adaptively selects optimizers and optimization hyperparameters throughout the optimization process accordingly. We leave this direction to future research.

## 7 CONCLUSION

In this work we developed and analyzed a Hessian eigenspace Lipschitz-aware minmax optimization algorithm `ELMO` by taking an eigendecomposition-centric approach to locally modelling a loss function. We then proved a widely applicable worst-case relative descent rate bound for quasi-Newton optimizers by comparison to `ELMO`. We experimented with the Lipschitz distributions, discovering that they are correlated with task difficulty and that they are helpful for optimizer and optimization hyperparameters selection — specifically, integrating second-order information into optimizers at the cost of additional computational complexity is worthwhile in settings where the convex Lipschitz parameters are small, but not those where they are large.

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

## A    QUASI-NEWTON CHALLENGES AND PROPOSED SOLUTIONS

Some of the challenges involved in training neural networks include:

- Because the models and data often have very complex structures, obtaining precisely optimal parameters is often computationally prohibitive. As a result, one must satisfy oneself with a small degree of suboptimality in the model's parameters, chosen to be small enough to satisfy one's needs while not exhausting the computational capacity at hand.

- Since many model architectures (e.g. artificial neural networks) have very complex structures, the loss function is generallhy non-convex as a function of the model's parameters. This makes finding the globally optimal choice of parameters an NP-hard problem (Pardalos & Vavasis, 1991; Manders & Adleman, 1978). As a result, one must satisfy oneself with merely a local minimum of the loss function (that is, a point at which the norm of the gradient w.r.t. the model parameters is zero, and the function is locally convex, or equivalently, the Hessian is positive semi-definite). This is often considered sufficient (see Soudry & Carmon (2016); Kawaguchi & Bengio (2019); Kawaguchi (2016); Nguyen et al. (2019)), however this does not apply to saddle points and local maxima (points at which the gradient norm is zero but the Hessian is not positive definite). Although some work has been done on trying to eliminate this problem by eliminating local- but not global- minima via overparameterization (Yu & Chen, 1995), further work (Ding et al., 2019) has shown that this does not scale to deep neural networks.

- As mentioned previously, there is no single universally optimal optimizer, even among existing optimizers.

As a result, sophisticated optimizers are necessary to contend with different neural network training scenarios. Restricting ourselves to quasi-newton optimization algorithms, scenarios with $n \gg 0$

(a common theme in machine learning, where $n$ may be in the millions, billions, or even trillions, as GPT4 (OpenAI et al., 2024) is rumored to have. See Patel & Wong (2023)) are that computing and inverting the Hessian (with respective complexities $\mathcal{O}\left(n^2\right), \mathcal{O}\left(n^3\right)$) may be computationally prohibitive. Also, one must ensure that

$$\Phi_t \succeq 0 \tag{11}$$

to ensure that $\theta_{t+1} - \theta_t$ is a descent direction of $f$. This is because $-\nabla f\left(\theta_t\right)$ is a descent direction of $f$, which implies that for all $v \in \mathbb{R}^n$, $-\alpha \cdot \nabla f\left(\theta_t\right)^\top v \cdot v^\top$ is a descent direction for $\alpha > 0$ and an ascent direction for $\alpha < 0$. However, if $(\lambda_i, v_i)$ is an eigenvalue-eigenvector pair of $\Phi_t$ with $\lambda_i < 0$ then $-v_i^\top \cdot \Phi_t \nabla f\left(\theta_t\right) \cdot v_i = -\lambda_i \nabla f\left(\theta_t\right)^\top \cdot v_i \cdot v_i$ which is an ascent direction, and then a better preconditioner could immediately be obtained by taking $\tilde{\Phi}_t$ with eigenpairs $\left(\tilde{\lambda}_j, \tilde{v}_j\right), \tilde{v}_j = v_j, \tilde{\lambda}_j = \begin{cases} \lambda_j & j \neq i \\ 0 & j = i \end{cases}$ to prevent an ascent in the subspace (a.k.a. eigenspace) $span(v_i)$.

Three common ways to contend with these challenges are:

- **The Hessian-Free approach** Making use of Pearlmutter (1994) to compute Hessian-vector products without explicit computation of the Hessian, one uses conjugate-gradient (Olver & Shakiban, 2006) iterations to compute progressively finer approximations to $\left(\mathcal{H}\left(\theta_t\right)\right)^{-1} \cdot \nabla f\left(\theta_t\right)$, stopping when one reaches a dimension with negative curvature. See, for instance, Martens (2010).

- **The Lanczos eigendecomposition approach** Making use of Lanczos iterations (Olver & Shakiban, 2006), one decomposes the Hessian into its eigendecomposition, and explicitly edits its eigenvalues. See, for instance, Dauphin et al. (2014); Sivan et al. (2024).

- **The Gauss-Newton approach** Using the generalized Gauss-Newton approximation to the Hessian (Esposito & Floudas, 2001; Schraudolph, 2002), one can obtain a matrix which has the following good properties:

  - Well approximated by a Kronecker product (sparse representation), which allows one to represent it and multiply by it very cheaply
  - Positive semi-definite
  - Can be computed with only a first-order loss function gradient oracle
  - Well approximates the true loss Hessian, when the second derivative of the model or the residual loss ($f\left(\theta_t\right) - f\left(\theta^*\right)$) is insignificant next to the generalized Gauss Newton

  Some examples of this approach include Agarwal et al. (2019); Botev et al. (2017); Gupta et al. (2018); Martens & Grosse (2015); Goldfarb et al. (2020); Anil et al. (2020). Of particular note are examples that make diagonal approximations to the Gauss-Newton, as noted by Martens (2020), that are most often viewed as first-order methods, such as Adagrad (Duchi et al., 2011a), RMSProp (Tieleman & Hinton, 2012), and Adam (Kingma & Ba, 2014). As noted by Martens (2020), due to the strong connection between the generalized Gauss-Newton and the Fischer Information matrix (when the loss function is cross-entropy loss (Good, 1952)), one can achieve certain theoretical benefits when using such methods, such as Fischer efficiency; see Amari (1998) for instance, which views $\theta_t$ as an unbiased estimator of $\theta^*$ of $f$, and uses the Cramer-Rao inequality (Jansen & Claeskens, 2011) to lower-bound the minimal number of iterations required to minimize the variance of said estimator as a function of the Fischer Information due to the number of samples consumed by each iteration.

See Nocedal & Wright (2006, Chapters 3.3,3.4) for further discussion of these approaches.

In order for a minimization problem to be well-defined, one must assume that $f$ is lower-bounded. We can infer from this that any subset of the domain space in which $f$ is concave must be a bounded set (because nonconstant concave functions with unbounded domains are not lower-bounded); this means that the second-order Taylor approximation of the function must have a bounded neighborhood in which it approximates the function well. Additionally, even in subsets of the domain space in which $f$ is convex, the neighborhood in which the second-order Taylor approximation of the loss function well-approximates the true loss function may be bounded. To address this, two common approaches been proposed in the literature, namely:

- **The Trust Regions Approach**, which explicitly maintains a radius of the neighborhood in which the second-order Taylor polynomial is a good approximation of the function, and bounds the step size to that radius. See Conn et al. (2000), Nocedal & Wright (2006, Chapter 4).

- **The Cubic Regularization Approach**, which assumes that Hessian is Lipschitz continuous (using the L2 vector-induced matrix norm to measure distances between Hessians), and as such can upper bound the distance between two points of the function using a third-order polynomial (discussed below, see Lemma 4.1.14 from Dennis & Schnabel (1983)). See Nesterov & Polyak (2006) for an algorithm based on this approach that adaptively estimates the Hessian-Lipschitz parameter.

## B  OTHER CONVERGENCE RATE MEASURES

**Convergence rates to first-order criticality**    Most works on convergence rates in the non-convex regime bound the number of iterations necessary to achieve first-order criticality ($\|\nabla f(\theta_t)\|_2 = 0$) by means of finding an $\epsilon_g$-stationary point (a point at which $\|\nabla f(\theta_t)\|_2 \le \epsilon_g$). The seminal work Wang et al. (2016) provide a convergence rate bound for general optimizers (with very weak assumptions) in the non-convex regime of $\mathcal{O}\left(\kappa^{\frac{2}{1-\nu}}(\Phi_t) \cdot \epsilon_g^{-\frac{1}{1-\nu}}\right)$ with learning rate $\alpha_t = \mathcal{O}\left(t^{-\nu}\right)$ and $\nu \in (0.5, 1)$. However, this bound is minimized by setting $\Phi_t$ to the minimizer of $\kappa(\Phi_t)$, which is a scalar matrix; this is equivalent to gradient descent, a first-order method. Experiments (see Sivan et al. (2024), for instance) and theory show that higher-order methods can achieve faster rates of convergence in our setting, demonstrating looseness of this convergence rate bound. See also D'efossez et al. (2020) who give such convergence rate bounds (requiring $t$ iterations, for $t$ s.t. $\frac{\sqrt{t}}{\log(t)} = \Omega\left(\epsilon_g^{-1}\right)$) for Adam and Adagrad, and Ward et al. (2019) who give such convergence rate bounds (at $\mathcal{O}\left(\epsilon_g^{-1}\right)$) for gradient descent with Adagrad-grafted step-sizes (see Agarwal et al. (2022) for a discussion on learning rate grafting).

**Convergence rates to second-order criticality**    A few go further in bounding the number of steps required to achieve second-order criticality (a point satisfying $\|\nabla f(\theta_t)\|_2 < \epsilon_g, -\lambda_{\min}(\mathcal{H}(\theta_t)) < \epsilon_H$). For instance, Nesterov & Polyak (2006); Cartis et al. (2011b); Xu et al. (2020) provide such bounds (at $\mathcal{O}\left(\max(\epsilon_g, \epsilon_H)^{-3}\right)$) on variants of the ARC algorithm, and Levy (2016); Jin et al. (2017); Ge et al. (2015) provide such bounds for varieties of SGD. This is of great importance since as noted, local minima are generally considered sufficiently optimal while local maxima/saddle points are not, despite being impossible to distinguish with only first-order criticality information. To the best of our knowledge, however, no such bounds exist in the general setting, nor do they even exist for the vast majority of existing optimization algorithms.

**Convergence rate dependence on preconditioner quality**    One possible quality metric for $\Phi_t$ is given by $\eta_t \triangleq \left\|\left(I - \mathcal{H}(\theta_t) \cdot \Phi_t\right) \cdot \frac{\nabla f(\theta_t)}{\|\nabla f(\theta_t)\|}\right\|_2$. In the convex regime, Nocedal & Wright (2006, Chapter 7.1) assume $\sup_t(\eta_t) < 1$ and prove that first-order criticality may be reached within $\mathcal{O}\left(\frac{\log \epsilon}{\log \frac{1+\sup_t(\eta_t)}{2}}\right)$ iterations. Adding an assumption of Lipschitz-continuity of the Hessian, they prove quadratic convergence to first-order criticality. O'Leary-Roseberry et al. (2019), in contrast, do not assume convexity but provide a bound on the parameter gap $\|\theta_t - \theta^*\|_2$ for $\eta_t$ satisfying the Eisenstat-Walker (Eisenstat & Walker, 1996; Dembo et al., 1982) condition $\eta_t \le \|\nabla f(\theta_t)\|_2$ on a Tikhonov-regularized Hessian. Like Wang et al. (2016), however, here too the constant in their bound is inversely proportional to $\zeta - \lambda_{\min}(\mathcal{H}(\theta_t))$ with $\zeta$ the Tikhonov regularization constant, thus is minimized by taking $\zeta \to \infty$, eliminating all second-order information and reverting to simple gradient descent. As before, this implies looseness due to the empirical success of making use of second-order methods.

## C  COMPARISON OF ELMO TO SELECT RELATED METHODS

ELMO is strikingly similar to Cauchy's method (not to be confused with Cauchy's Steepest Descent method (Nocedal & Wright, 2006, Chapter 4.1)) and Newton's method mentioned above. In this section, we note the similarity between them, and the sources of the differences between them.

### C.1  COMPARISON TO CAUCHY'S METHOD

Cauchy's method (Traub, 1982) is nearly identical to ELMO:

$$
(\theta_{t+1} - \theta_t)^\top_{cauchy} \cdot v_i \triangleq -\frac{2}{1 + \sqrt{1 - 2\frac{L_t^i \cdot \nabla f(\theta_t)^\top v_i}{\lambda_i^2(\theta_t)}}} \cdot \frac{\nabla f(\theta_t)^\top v_i}{\lambda_i(\theta_t)}
$$

$$
= -\frac{2}{1 + \frac{1}{|\lambda_i(\theta_t)|} \cdot \sqrt{\lambda_i^2(\theta_t) - 2L_t^i \cdot \nabla f(\theta_t)^\top v_i}} \frac{\nabla f(\theta_t)^\top v_i}{\lambda_i(\theta_t)}
$$

$$
= \frac{-2\nabla f(\theta_t)^\top v_i}{\lambda_i(\theta_t) + \frac{\lambda_i(\theta_t)}{|\lambda_i(\theta_t)|} \cdot \sqrt{\lambda_i^2(\theta_t) - 2L_t^i \cdot \nabla f(\theta_t)^\top v_i}}
$$

$$
= \frac{1}{L_t^i} \cdot \frac{-2L_t^i \cdot \nabla f(\theta_t)^\top v_i}{\lambda_i(\theta_t) + \frac{\lambda_i(\theta_t)}{|\lambda_i(\theta_t)|} \cdot \sqrt{\lambda_i^2(\theta_t) - 2L_t^i \cdot \nabla f(\theta_t)^\top v_i}}
$$

$$
= \frac{1}{L_t^i} \cdot \frac{\left(\lambda_i^2(\theta_t) - 2L_t^i \cdot \nabla f(\theta_t)^\top v_i\right) - \lambda_i^2(\theta_t)}{\lambda_i(\theta_t) + \frac{\lambda_i(\theta_t)}{|\lambda_i(\theta_t)|} \cdot \sqrt{\lambda_i^2(\theta_t) - 2L_t^i \cdot \nabla f(\theta_t)^\top v_i}}
$$

$$
= \frac{\sqrt{\lambda_i^2(\theta_t) - 2L_t^i \cdot \nabla f(\theta_t)^\top v_i} - \sqrt{\lambda_i^2(\theta_t)}}{L_t^i} \cdot \frac{|\lambda_i(\theta_t)|}{\lambda_i(\theta_t)}
$$

$$
= \frac{\sqrt{\lambda_i^2(\theta_t) - 2L_t^i \cdot \nabla f(\theta_t)^\top v_i} - \frac{|\lambda_i(\theta_t)|}{\lambda_i(\theta_t)} \cdot \lambda_i(\theta_t)}{L_t^i} \cdot \frac{|\lambda_i(\theta_t)|}{\lambda_i(\theta_t)}
$$

$$
= \frac{\frac{|\lambda_i(\theta_t)|}{\lambda_i(\theta_t)} \cdot \sqrt{\lambda_i^2(\theta_t) - 2L_t^i \cdot \nabla f(\theta_t)^\top v_i} - \lambda_i(\theta_t)}{L_t^i}
$$

The difference between our minimization step and their step is merely the sign on the squareroot. The difference lies in removing the absolute value in equation 3's 3rd-order term and taking the negative root of its derivative, due to the difference in goals: we attempt to minimize the function, leading us to select the positive step. They attempt to find the function's critical points, leading them to select the negative step.

### C.2  COMPARISON TO NEWTON'S METHOD

Unlike Cauchy's method, Newton's method (in optimization) makes a second-order approximation to the function's gradient. This is equivalent to the Hessian being constant, which is equivalent to $L_H = 0$. Indeed, taking the limit of equation 7 when $L_H \to 0^+$, we recover Newton's method:

$$
\lim_{L_H \to 0^+} \Delta\theta_t^{*\top} v_i = \lim_{L_H \to 0^+} -\frac{2\nabla f(\theta_t)^\top v_i}{\sqrt{\lambda_i^2(\theta_t) - 2L_t^i \cdot \nabla f(\theta_t)^\top v_i} + \lambda_i(\theta_t)}
$$

$$
= \begin{cases} -\frac{\nabla f(\theta_t)^\top v_i}{\lambda_i} & \lambda_i > 0 \\ \infty & \lambda_i < 0 \end{cases}
\tag{12}
$$

## D  CONVERGENCE RATE DEPENDENCE ON HESSIAN-LIPSCHITZ PARAMETER

As noted by Griewank (1981), the Hessian-Lipschitz parameter (in our case, the respective constants of each eigenspace) may be computationally difficult to obtain precisely, leading some optimization algorithms to estimate it approximately instead of computing it precisely (e.g. ARC). In order to balance the computational burden of computing it to a high degree of exactitude with the degradation of an algorithm's convergence rate that comes with poor estimations, we study the effects of the Hessian-Lipschitz parameter on $M_t^i\left(\Delta\theta_t^\top v_i\right)$.

### D.1  LIPSCHITZ ROBUSTNESS

To address the convergence rate's robustness to overly conservative $L_t^i$, we consider the case when $L_t^i \to \infty$.

**Theorem D.1.** *Let $f : \mathbb{R}^n \to \mathbb{R}$ satisfying assumptions 1 and 2. Then*

$$M_t^i\left(\Delta\theta_t^{*\top} v_i\right) = \Theta\left(\frac{1}{\sqrt{L_t^i}}\right)$$

*when $L_t^i \to \infty$*

*Proof.*

$$\lim_{L_t^i \to \infty} \frac{M_t^i\left(\Delta\theta_t^{*\top} v_i\right)}{\frac{-12\left(-\nabla f(\theta_t)^\top v_i\right)^{1.5} - 36\left(-\nabla f(\theta_t)^\top v_i\right) + \sqrt{2}\sqrt{-\nabla f(\theta_t)^\top v_i} + 3\sqrt{2}}{\frac{3}{-\nabla f(\theta_t)^\top v_i} - 18\sqrt{2}} \cdot \frac{1}{\sqrt{L_t^i}}}$$

$$= \lim_{L_t^i \to \infty} -\frac{1}{\sqrt{\frac{\lambda_i^2(\theta_t)}{L_t^i\left(-2\cdot\nabla f(\theta_t)^\top v_i\right)} + 1} + \frac{\lambda_i(\theta_t)}{\sqrt{L_t^i}\sqrt{-2\cdot\nabla f(\theta_t)^\top v_i}}}$$

$$\cdot \frac{6\sqrt{-\nabla f\left(\theta_t\right)^\top v_i}}{6\sqrt{-\nabla f\left(\theta_t\right)^\top v_i} - 2\cdot\nabla f\left(\theta_t\right)^\top v_i}$$

$$- \frac{2\cdot\nabla f\left(\theta_t\right)^\top v_i}{6\sqrt{-\nabla f\left(\theta_t\right)^\top v_i} - 2\cdot\nabla f\left(\theta_t\right)^\top v_i}$$

$$\cdot\left(\sqrt{\frac{\lambda_i^2\left(\theta_t\right)}{L_t^i\left(-2\cdot\nabla f\left(\theta_t\right)^\top v_i\right)} + 1} - \frac{\lambda_i\left(\theta_t\right)}{\sqrt{L_t^i}\sqrt{-2\cdot\nabla f\left(\theta_t\right)^\top v_i}}\right)^3$$

$$+ \frac{1}{\sqrt{L_t^i}} \cdot \frac{1 + 6\sqrt{2}\nabla f\left(\theta_t\right)^\top v_i}{2\left(6\sqrt{-\nabla f\left(\theta_t\right)^\top v_i} - 2\cdot\nabla f\left(\theta_t\right)^\top v_i\right)} \frac{\lambda_i\left(\theta_t\right)\left(\frac{-2\cdot\nabla f(\theta_t)^\top v_i}{\sqrt{\frac{\lambda_i^2(\theta_t)}{L_t^i} - 2\cdot\nabla f(\theta_t)^\top v_i} + \frac{\lambda_i(\theta_t)}{\sqrt{L_t^i}}}}\right)^2}{\left(\frac{1}{6} + \sqrt{2}\nabla f\left(\theta_t\right)^\top v_i\right)\sqrt{-2\cdot\nabla f\left(\theta_t\right)^\top v_i}}$$

$$= 1$$

$$\square$$

### D.2  BENEFIT OF LIPSCHITZ TIGHTNESS

To see how minimizing $L_t^i$ as much as possible benefits the bound, we consider the case when $L_t^i \to 0^+$.

**Theorem D.2.** *Benefit of Lipschitz tightness: concave subspaces*

*Let $f : \mathbb{R}^n \to \mathbb{R}$ satisfying assumptions 1 and 2. If $\lambda_i(\theta_t) \leq 0$ then*

$$M_t^i \left( \Delta \theta_t^{*\top} v_i \right) = \Theta \left( -\frac{1}{L_t^{i,2}} \right)$$

*when $L_t^i \to 0^+$*

*Proof.*

$$\lim_{L_t^i \to 0^+} \frac{M_t^i \left( \Delta \theta_t^{*\top} v_i \right)}{\frac{2}{3} \frac{\lambda_i^3(\theta_t)}{L_t^{i,2}}}$$

$$= \lim_{L_t^i \to 0^+} \left( 3 \left( \frac{\sqrt{1 - 2L_t^i \cdot \frac{\nabla f(\theta_t)^\top v_i}{(-\lambda_i(\theta_t))^2}} + 1}{2} \right)^2 - 2 \left( \frac{\sqrt{1 - 2L_t^i \cdot \frac{\nabla f(\theta_t)^\top v_i}{(-\lambda_i(\theta_t))^2}} + 1}{2} \right)^3 \right)$$

$$+ L_t^i \cdot \frac{3}{2\lambda_i^3(\theta_t)} \cdot \left( \sqrt{\lambda_i^2(\theta_t) - 2L_t^i \cdot \nabla f(\theta_t)^\top v_i} - \lambda_i(\theta_t) \right) \cdot \nabla f(\theta_t)^\top v_i$$

$$= 1$$

$\square$

**Theorem D.3.** *Benefit of Lipschitz tightness: convex subspaces*

*Let $f : \mathbb{R}^n \to \mathbb{R}$ satisfying assumptions 1 and 2. If $\lambda_i(\theta_t) > 0$ then*

$$M_t^i \left( \Delta \theta_t^{*\top} v_i \right) - M_t^i \left( \lim_{L_t^i \to 0^+} \Delta \theta_t^{*\top} v_i \right) = \Theta \left( L_t^i \right)$$

*when $L_t^i \to 0^+$*

*Proof.* We begin by noting that by equation 12, $\lim_{L_t^i \to 0^+} \Delta \theta_t^{*\top} v_i = \frac{\left| \nabla f(\theta_t)^\top v_i \right|}{\lambda_i(\theta_t)}$. Plugging this into $M_t^i$:

$$\lim_{L_t^i \to 0^+} \frac{M_t^i \left(\Delta\theta_t^{*\top} v_i\right) - M_t^i \left(\lim_{L_t^i \to 0^+} \Delta\theta_t^{*\top} v_i\right)}{\frac{1}{2}\lambda_i(\theta_t) \cdot \frac{\left(\nabla f(\theta_t)^\top v_i\right)^3}{\lambda_i^4(\theta_t)} - \frac{\left(\nabla f(\theta_t)^\top v_i\right)^3}{2\lambda_i^2(\theta_t)}}$$

$$= \lim_{L_t^i \to 0^+} -\frac{\left(\nabla f(\theta_t)^\top v_i\right)^3}{2\lambda_i^2(\theta_t)} \cdot \left(\frac{4}{\left(\sqrt{1 - 2L_t^i \cdot \frac{\nabla f(\theta_t)^\top v_i}{\lambda_i(\theta_t)}} + 1\right)^2}\right)$$

$$+ \frac{1}{2}\lambda_i(\theta_t) \cdot \frac{\left(\nabla f(\theta_t)^\top v_i\right)^3}{\lambda_i^4(\theta_t)} \cdot \left(\frac{\frac{2\lambda_i(\theta_t)}{\sqrt{\lambda_i^2(\theta_t) - 2L_t^i \cdot \nabla f(\theta_t)^\top v_i + \lambda_i(\theta_t)}} + 1}{2}\right)$$

$$\cdot \left(\frac{2\lambda_i^2(\theta_t)}{\lambda_i(\theta_t)\sqrt{\lambda_i^2(\theta_t) - 2L_t^i \cdot \nabla f(\theta_t)^\top v_i + \lambda_i^2(\theta_t)}}\right)\left(\frac{2}{1 + \sqrt{1 - 2L_t^i \cdot \frac{\nabla f(\theta_t)^\top v_i}{\lambda_i^2(\theta_t)}}}\right)$$

$$- \frac{L_t^i}{4} \cdot \frac{\left(\nabla f(\theta_t)^\top v_i\right)^4}{\lambda_i^5(\theta_t)} \left(\frac{2}{\sqrt{1 - 2L_t^i \cdot \frac{\nabla f(\theta_t)^\top v_i}{\lambda_i^2(\theta_t)}} + 1}\right)\left(\frac{2}{\sqrt{1 - 2L_t^i \cdot \frac{\nabla f(\theta_t)^\top v_i}{\lambda_i^2(\theta_t)}} + 1}\right)$$

$$\cdot \left(\frac{1 + \frac{2}{\sqrt{1 - 2L_t^i \cdot \frac{\nabla f(\theta_t)^\top v_i}{\lambda_i^2(\theta_t)}} + 1} + \frac{4}{\left(\sqrt{1 - 2L_t^i \cdot \frac{\nabla f(\theta_t)^\top v_i}{\lambda_i^2(\theta_t)}} + 1\right)^2}}{3}\right)$$

$$= 1$$

$\square$

## E    EVIDENCE FROM THE LITERATURE

Experiments by Alain et al. (2018); Sagun et al. (2016); Ghorbani et al. (2019); Gur-Ari et al. (2018) show that the positive eigenvalues of the Hessian remain relatively stable throughout training, while the negative eigenvalues shrink rapidly. Alain et al. (2018) and Sagun et al. (2016) also show that the negative eigenvalues shift chaotically. Gur-Ari et al. (2018) show that when training a network on a classification task with $k$ classes, then at least the eigenspace spanned by the $k$ eigenvectors corresponding to the top $k$ eigenvalues remains very stable. Sivan et al. (2024); Liu et al. (2024) also show that when training a neural network on a variety of tasks, the top $k$ eigenvalues and their corresponding eigenvectors change very slowly. Alain et al. (2018) also show explicitly that the second-order Taylor approximation is a poor approximation of the loss function in the eigenspace corresponding to the negative eigenvalues (the concave eigenspace), but an excellent approximation in the eigenspace corresponding to the positive eigenvalues (the convex eigenspace); indeed, they show that the optimal step in the convex eigenspace is well estimated by the Newton step, while there is no correlation between the Hessian and the optimal step in the concave eigenspace. Using the Lipschitz parameter as a measure of the rate of change of the Hessian in a given subspace (hence a measure of the quality of a second-order Taylor approximation to a function and its corresponding Newton step), this supports the claim that $L^+ \ll L^-$.

## F    LIPSCHITZ PARAMETER EXPERIMENTS

We tested our algorithm in 7 scenarios with the PyTorch 1.13.0 framework, each on a single NVIDIA GeForce RTX 3090 GPU with the standard hyperparameters and settings for ARC:

- $\sigma_0^+ = \sigma_0^- = 1$
- $\eta_1 = 0.1, \eta_2 = 0.9$
- $\gamma_1 = \gamma_2 = 2$
- Maximum sub-problem failures = 11
- Maximum sub-problem iterations = 50,
- Sub-problem tolerance = $10^{-6}$
- Lanczos eigendecomposition (Garber et al., 2016)
- BFGS trinomial sub-problem solver (Nocedal & Wright, 2006, Chapter 6.1), (Feinman, 2021)
- trinomial sub-problem maximal failures=11
- trinomial sub-problem maximal iterations=50

The test scenarios[1] include combinations of:

- Training ResNet18 artificial neural networks (He et al., 2015) for image classification on MNIST, CIFAR10, and FakeData (random noise in place of images) with Cross-Entropy Loss (CELoss), to evaluate the effect of changing data on the Lipschitz parameters
- Training a CNN for image classification on CIFAR10 with CELoss, to evaluate the effect of changing neural network architecture on the Lipschitz parameters.
- Training a CNN[2] autoencoder[3] (LeCun, 1987) to compress MNIST

The classification CNN architecture:

1. A feature extractor consisting of 2 2D convolutional layers with 6 output channels for the first and 16 output channels for the second. Both had kernel sizes of 5 pixels. Each of these is followed by a ReLU nonlinearity and then 2x2 2D max pooling
2. A 3-layer MLP classification head with hidden sizes (120, 84) and ReLU nonlinearities

The autoencoder CNN architecture:

- Encoder: 4 2D convolutional layers with respective output channel numbers (16,32,32,64) and kernel sizes of 3 pixels for the first two and 5 pixels for the last two. The first two have 1 pixel padding and the last two have 2 pixel paddings. After every layer we apply LeakyReLU nonlinearity and after every 2 layers we apply 2x2 2D max pooling.
- Decoder: 4 composite layers consisting of
    1. a 2D transpose-convolutional layer
    2. LeakyReLU nonlinearity (only for first and third composite layers)
    3. a 2d convolutional layer
    4. LeakyReLU nonlinearity
- Decoder hidden channel sizes: 32-32-16-16-16-16-3
- Decoder kernel sizes: 2-5-5-5-2-3-5-3
- All strides are of size 1, except for the first and third transpose-convolutional layers of the decoder, with stride of size 2
- Decoder paddings: 0-2-2-2-0-1-2-1

Each experiment took several hours to run. All experiments (including those from above) shown in figure 3.

---

[1]Code available on Github at REDACTED

[2]convolutional neural network

[3]With hidden dimensions 128-64-36-18-9-18-36-64-128, ReLU nonlinearities, and sigmoid nonlinearity on the output

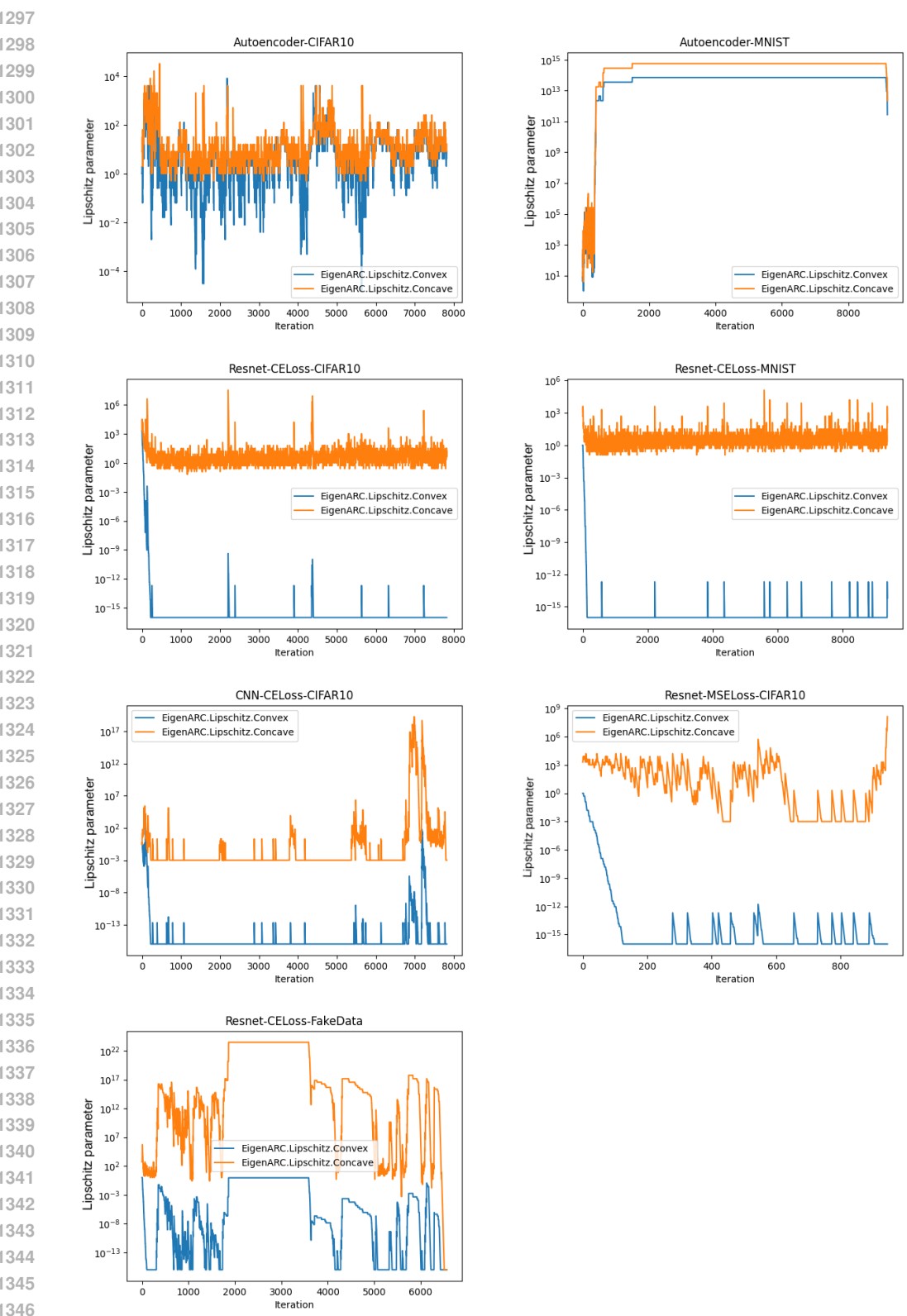

Figure 3: Comparisons of convex-subspace Lipschitz parameters to concave-subspace Lipschitz parameters. *Logarithmic scale*

One caveat is that due to computational constraints, we use stochastic minibatch training for the neural networks instead of using the full batch to compute the gradient and Hessian-vector products at each iteration (see Bertsekas (1996) for an introduction to minibatch Monte-Carlo estimation of a sum). However, Cartis et al. (2011a), notes that the adaptive Lipschitz parameter estimates may account for this variance by being greater than the actual Lipschitz parameters. Thus, our claims of $L^+ \ll 1$ are not affected (since our experiments effectively provide an upper bound on $L^+$) while our claims of $L^- \gg 0$ are weakened. Since there is no reason to expect the variance on $L^-$ to be significantly greater than the variance on $L^+$, however, our experiments remain valid.

For visual clarity, the quasi-Newton superiority measurements in 2 are presented after:

1. Clipping extreme values to the 10% - 90% quantile range
2. Gaussian smoothing, consisting of a rolling window of size 300 and standard deviation of 100

### F.1 COMPUTATION OF LIPSCHITZ PARAMETERS

We modified the standard ARC algorithm to compute distinct Lipschitz parameters for the eigenspaces corresponding to the minimal and maximal eigenvalues. Pseudocode for this algorithm is given below.

---

**Algorithm 2** Algorithm `EigenARC`

---

**Require:** $\epsilon \in \mathbb{R}^+, \theta_0 \in \mathbb{R}^n, \gamma_1 > 1 > \gamma_2 > 0, \eta_2 \geq \eta_1 > 0, \left(L_0^i\right)_{i=1}^n > 0, \texttt{EIGEN}, \texttt{BASE\_OPT}$
1:   $t \leftarrow 0$
2:   **while** $\|\nabla f(\theta_t)\|_2 > \epsilon$ **do**           ▷ While `BASE_OPT` hasn't converged yet
3:      $(\lambda_i, v_i)_{i=1}^n \leftarrow \texttt{EIGEN}(\mathcal{H}(\theta_\mathtt{t}))$
4:      **if** $\texttt{ASSESS\_LIPSCHITZ}\left((\mathtt{L_t^i})_{\mathtt{i=1}}^\mathtt{n}\right) > \eta_2$ **then**          ▷ Overly conservative $L_t^i$
5:         $\left(L_t^i\right)_{i=1}^n \leftarrow \gamma_2 \cdot \left(L_t^i\right)_{i=1}^n$
6:      **else**
7:         **if** $\texttt{ASSESS\_LIPSCHITZ}\left((\mathtt{L_t^i})_{\mathtt{i=1}}^\mathtt{n}\right) < \eta_1$ **then**       ▷ Overly liberal $L_t^i$
8:            **while** $\texttt{ASSESS\_LIPSCHITZ}\left((\mathtt{L_t^i})_{\mathtt{i=1}}^\mathtt{n}\right) > \eta_1$ **do**   ▷ Raise all $L_t^i$ assessment is passed
9:               $\left(L_t^i\right)_{i=1}^n \leftarrow \gamma_1 \cdot \left(L_t^i\right)_{i=1}^n$
10:           **end while**
11:           **for** i=1,...,n **do**   ▷ Reduce the $L_t^i$ that can be reduced without violating assessment
12:             **while** $\texttt{ASSESS\_LIPSCHITZ}\left((\mathtt{L_t^i})_{\mathtt{i=1}}^\mathtt{n}\right) > \eta_1$ **do**
13:                $L_t^i \leftarrow \frac{L_t^i}{\gamma_1}$
14:             **end while**
15:                $L_t^i \leftarrow \gamma_1 \cdot L_t^i$
16:           **end for**
17:         **end if**
18:      **end if**
19:      $\theta_{t+1} \leftarrow \texttt{BASE\_OPT}(\theta_\mathtt{t})$
20:      $t \leftarrow t + 1$
21: **end while**
**return** $\left(L_{\hat{t}}^i\right)_{i=1, \hat{t}=1}^{n, t}$

   **procedure** ASSESS_LIPSCHITZ($\left(\hat{L}_t^i\right)_{i=1}^n$)
      **return** $\dfrac{f(\theta_t) - f\left(\theta_t + \sum_{i=1}^n \Delta\theta_t^{*\top} v_i\left(\hat{L}_t^i\right) \cdot v_i\right)}{-\sum_{i=1}^n M_t^i\left(\Delta\theta_t^{*\top} v_i\left(\hat{L}_t^i\right)\right)}$
   **end procedure**

---

While lines 3-18 of `EigenARC` may technically be usable as the `LIPSCHITZ` subroutine of algorithm `ELMO` above, each iteration requires $\Omega(n)$ evaluations of the loss function, which will be computationally expensive if $n \gg 0$ and if the loss function is computationally heavy. This may be ameliorated by performing these calculations only for a small subset of the eigenspaces like Sivan et al. (2024), however we leave this to future work.

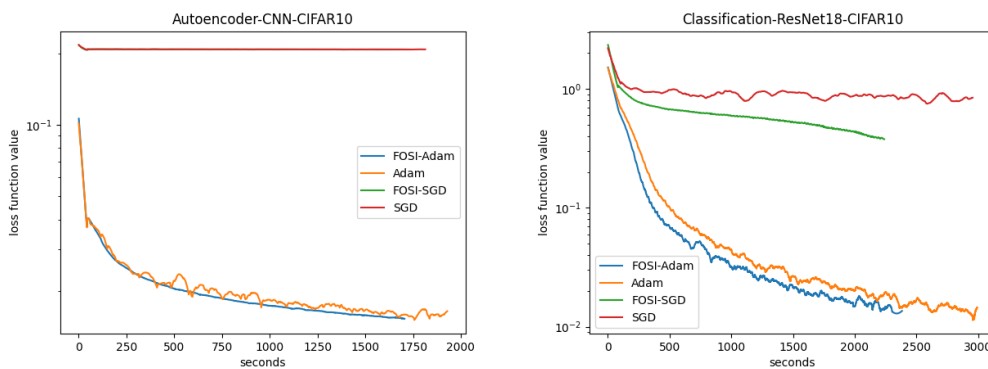

(a) The setting in which we train a CNN on an autoencoder task has large convex Lipschitz parameters throughout training

(b) The setting in which we train ResNet18 on a classification task has small convex Lipschitz parameters throughout training

Figure 4: Comparison of second-order optimizers against first-order optimizers in settings with different sized convex Lipschitz parameters. Second-order optimizers only hold an advantage over first-order optimizers (thus justifying their additional computational complexity) when the convex Lipschitz parameters are small.

## G  LIPSCHITZ-AIDED OPTIMIZER SELECTION

In this section, we demonstrate the use of convex Lipschitz parameters to select the best optimizer for our use case. Due to the relative constancy of Lipschitz parameters throughout the training process (after an initial warmup phase) in different settings, we can select optimizers for each setting based on the following rule: **quasi-Newton optimizers hold an advantage over first-order optimizers when the convex Lipschitz parameters are small**. As discussed in section 5, the convex Lipschitz parameters in the image autoencoder training setting are far larger than those in the image classification setting, so we compare a quasi-Newton optimizer against first-order methods in these settings to validate our rule.

FOSI Sivan et al. (2024) is a variant of Saddle-Free Newton Dauphin et al. (2014) which applies Newton iterations in the domain space subspaces spanned by the dominant eigenvectors of the Hessian, and a first-order "base optimizer" in the remaining subspaces. We use FOSI as our representative second-order optimizer due to its computational effectiveness, capability to adjust the computational complexity of each iteration by adjusting the number of "dominant" eigenvectors to compute (fewer eigenvectors comes at the cost of a poorer Hessian approximation by approximating the Hessian with a lower-rank matrix, although this is somewhat mitigated by applying the base optimizer in these subspaces), and fairness of comparison (since its integration of first-order optimizers allows us to compare the effect of second-order optimization in the dominant eigenspaces against first-order optimization in these spaces, while all else is held equal - the remaining subspaces are both treated by the same first-order optimizers).

Experiment results may be seen in figure 4.

The experiments are run with the same settings as before, with FOSI augmenting SGD and Adam respectively and Savitzky-Golay order-2 filtering with a window size of 5000 for clarity of visualization. It may be clearly seen that FOSI second-order augmentation is beneficial only in the classification setting, due to the small convex Lipschitz parameters.

## H  PROOFS

### H.1  PRELIMINARY LEMMAS

Before we can get started, we prove a few basic lemmas.

**Lemma H.1.**
$$\forall_{x \geq -1} : \sqrt{1+x} \leq 1 + \frac{x}{2}$$

*Proof.* Mark $g : \mathbb{R} \to \mathbb{R}, g(x) = 1 + \frac{x}{2} - \sqrt{1+x}$. We have

$g'(x) = \frac{1}{2}\left(1 - \frac{1}{\sqrt{1+x}}\right)$

$g''(x) = \frac{1}{4}\left(\frac{1}{(1+x)^{\frac{3}{2}}}\right)$

$g$ is convex due to its second derivative being positive for all $x > -1$. Therefore, its sole critical point $x = 0$ obtained from the derivative is a minimum, and $\forall_{x \geq -1} : g(x) \geq g(0) = 0$ $\square$

**Corollary H.1.1.**
$$\forall_{x \in \mathbb{R}^+} \forall_{y \geq -x} : \sqrt{x+y} \leq \sqrt{x} + \frac{y}{2\sqrt{x}}$$

*Proof.*
$$\sqrt{x+y} = \sqrt{x}\sqrt{1 + \frac{y}{x}} \leq \sqrt{x}\left(1 + \frac{y}{2x}\right) = \sqrt{x} + \frac{y}{2\sqrt{x}}$$

$\square$

**Lemma.** *Let $f : \mathbb{R}^n \to \mathbb{R}$ satisfy assumptions 1 and 2. Then*
$$m_t^i\left(\Delta\theta_t^{*\top} v_i\right) \leq M_t^i\left(\Delta\theta_t^{*\top} v_i\right) \leq 0 \tag{13}$$

*Proof.* The first inequality stems from the trivial fact that $m_t^i \leq M_t^i$.

The second inequality follows from the fact that (by design), $\Delta\theta_t^{*\top} v_i$ is a minimizer of $M_t^i\left(\Delta\theta_t^\top v_i\right)$, but
$$M_t^i(0) = 0$$

$\square$

**Lemma.** *4.1 Minmax preconditioner*

$$\underset{\Phi_t \in \mathbb{R}^{n \times n}}{\arg\min}\left|\Delta\Delta^i\theta_t(\Phi_t)\right| = \left(\frac{\mathcal{H}(\theta_t) + \sqrt{(\mathcal{H}(\theta_t))^2 + 2V \cdot diag\left(L_t^i \cdot \left|\nabla f(\theta_t)^\top v_i\right|\right)_{i=1}^n \cdot V^\top}}{2}\right)^{-1}$$

*Proof.*

$$\left|\Delta\Delta^i\theta_t\right| = \left|\nabla f(\theta_t)^\top\left(\Phi_t - \frac{2}{\lambda_i + \sqrt{\lambda_i^2 - 2L_t^i \cdot \nabla f(\theta_t)^\top v_i}}I\right) v_i\right|$$

$$= \left|\nabla f(\theta_t)^\top\left(\Phi_t - \left(\frac{\mathcal{H}(\theta_t) + \sqrt{(\mathcal{H}(\theta_t))^2 + 2L_t^i \cdot \left|\nabla f(\theta_t)^\top v_i\right| I}}{2}\right)^{-1}\right) v_i\right|$$

and the result follows from developing the second parenthesized term for all $n$ dimensions of the domain space. $\square$

## H.2 LEMMA 3.1: EIGENSPACE DESCENT

Working with assumptions 1 and equation 2, Dennis & Schnabel (1983, Lemma 4.1.14) prove the following:

**Lemma H.2.**

$$f\left(\theta_{t+1}\right) - f\left(\theta_t\right) \leq \nabla f\left(\theta_t\right)^T \cdot \Delta\theta_t + \frac{1}{2}\Delta\theta_t^T \mathcal{H}\left(\theta_t\right)\Delta\theta_t + \frac{1}{6}L_H \left\|\Delta\theta_t\right\|_2^3 \tag{14}$$

Much like algorithm `ELMO`, minimizing equation 14 would maximize [a bound on] the descent given by iteration $t$. However, previous works such as Nesterov & Polyak (2006) note the difficulty of minimizing this 3rd-order $n$-dimensional polynomial, even when $L_H$ is known. Indeed, Cartis et al. (2011a) propose minimizing it iteratively over a growing subspace, with each iteration's minimization subspace a superset of the previous iterations' (in practice, they use the Hessian's Krylov subspaces, initialized with the gradient). In our theoretical analysis however, we have the freedom to simply take the most natural decomposition of the space into subspaces, the eigenspaces of the Hessian. This does not limit the practicality of our approach, however, since Lanczos methods allow one to obtain elements of this decomposition. In fact, Sivan et al. (2024) demonstrate experimentally that decomposing the parameter space into multiple eigenspaces and optimizing each separately can significantly speed up optimization wall time, despite the additional computational burden of the Lanczos iterations, because of the regularizing effect this has on the function in each of the subspaces (by reducing the variance of the Hessian eigenvalues). Ghorbani et al. (2019) also show the benefits of reducing this variance.

**Lemma.** *3.1 Eigenspace Descent Bounds*

*Let $f : \mathbb{R}^n \to \mathbb{R}$ be a function satisfying assumptions 1 and 2, and let $\theta_{t+1} \in \mathbb{R}^n$. Marking $\Delta\theta_t = \theta_{t+1} - \theta_t$, we have*

$$\exists_{\left(L_t^i\right)_{i=1}^n \in (\mathbb{R}^+)^n} : f\left(\theta_{t+1}\right) - f\left(\theta_t\right) \leq \sum_{i=1}^n M_t^i \left(\Delta\theta_t^\top v_i\right) \tag{15}$$

$$\exists_{\left(L_t^i\right)_{i=1}^n \in (\mathbb{R}^+)^n} : f\left(\theta_{t+1}\right) - f\left(\theta_t\right) \geq \sum_{i=1}^n m_t^i \left(\Delta\theta_t^\top v_i\right) \tag{16}$$

We give 2 proofs of the above lemma. The first proof is far simpler and relies on the standard spectral norm-Lipschitz continuous Hessian assumption given by equation 2 instead of assumption 2:

*Proof.* Beginning with Nesterov & Polyak (2006, Lemma 1) for the first inequality,

$$\left| f\left(\theta_{t+1}\right) - f\left(\theta_t\right) - \left(\nabla f\left(\theta_t\right)^\top \left(\theta_{t+1} - \theta_t\right) + \left(\theta_{t+1} - \theta_t\right)^\top \mathcal{H}\left(\theta_t\right)\left(\theta_{t+1} - \theta_t\right)\right) \right|$$
$$\leq L_H \left\| \theta_{t+1} - \theta_t \right\|_2^3$$

$$\left| f\left(\theta_{t+1}\right) - f\left(\theta_t\right) - \left( \sum_{i=1}^n \left( \nabla f\left(\theta_t\right)^\top v_i \cdot \left(\theta_{t+1} - \theta_t\right)^\top v_i + \lambda_i \left( \left(\theta_{t+1} - \theta_t\right)^\top v_i \right)^2 \right) \right) \right|$$

$$\leq L_H \left\| \sum_{i=1}^n \left(\theta_{t+1} - \theta_t\right)^\top v_i v_i^\top \right\|_2^3$$

$$\leq L_H \left( \sum_{i=1}^n \left\| \left(\theta_{t+1} - \theta_t\right)^\top v_i v_i^\top \right\|_2 \right)^3$$

$$= L_H \cdot n^3 \left( \sum_{i=1}^n \frac{1}{n} \left| \left(\theta_{t+1} - \theta_t\right)^\top v_i \right| \right)^3$$

$$\leq L_H \cdot n^3 \left( \frac{1}{n} \sum_{i=1}^n \left| \left(\theta_{t+1} - \theta_t\right)^\top v_i \right|^3 \right)$$

$$= L_H \cdot n^2 \sum_{i=1}^n \left| \left(\theta_{t+1} - \theta_t\right)^\top v_i \right|^3$$

$$= \sum_{i=1}^n \tilde{L_H} \cdot \left| \left(\theta_{t+1} - \theta_t\right)^\top v_i \right|^3$$

with

1. the second inequality being a representation of $\theta_{t+1} - \theta_t$ over the (orthogonal) Hessian eigenbasis

2. the third inequality due to the triangle inequality

3. the fourth inequality due to Jensen's inequality

$\square$

Our first proof of lemma 3.1 is simple, but leaves something to be desired due to its lack of per-eigenspace Lipschitz parameters and due to the presence of $n^2$ in the bound, which can be very large, as noted in section A. The first proof's assumption of equation 2 is also easily seen to be no weaker than assumption 2 (meaning that assuming equation 2 implies assumption 2) by taking $L_R \triangleq L_t^i \triangleq L_H$ for all $t \in [T], i \in [n]$. To address these concerns, we make use of assumption 2, and give a second (though more complicated) proof for lemma 3.1. We'll prove only the upper bound, as a proof for the lower bound is similar.

*Proof.*

$$f\left(\theta_{t+1}\right) - f\left(\theta_t\right)$$

$$= \int_0^1 \nabla f\left(\theta_t + y\left(\theta_{t+1} - \theta_t\right)\right)^\top \left(\theta_{t+1} - \theta_t\right) dy$$

$$= \nabla f\left(\theta_t\right)^\top \left(\theta_{t+1} - \theta_t\right) + \int_0^1 \left(\nabla f\left(\theta_t + y\left(\theta_{t+1} - \theta_t\right)\right) - \nabla f\left(\theta_t\right)\right)^\top \left(\theta_{t+1} - \theta_t\right) dy$$

$$= \nabla f\left(\theta_t\right)^\top \left(\theta_{t+1} - \theta_t\right) + \int_0^1 \left(\int_0^1 y \mathcal{H}\left(\theta_t + yz\left(\theta_{t+1} - \theta_t\right)\right)\left(\theta_{t+1} - \theta_t\right) dz\right)^\top \left(\theta_{t+1} - \theta_t\right) dy$$

$$= \nabla f\left(\theta_t\right)^\top \left(\theta_{t+1} - \theta_t\right) + \int_0^1 \int_0^1 y \left(\theta_{t+1} - \theta_t\right)^\top \mathcal{H}\left(\theta_t + yz\left(\theta_{t+1} - \theta_t\right)\right)\left(\theta_{t+1} - \theta_t\right) dydz$$

$$= \nabla f\left(\theta_t\right)^\top \left(\theta_{t+1} - \theta_t\right) + \left(\theta_{t+1} - \theta_t\right)^\top \mathcal{H}\left(\theta_t\right)\left(\theta_{t+1} - \theta_t\right)$$

$$+ \underbrace{\int_0^1 \int_0^1 y \left(\theta_{t+1} - \theta_t\right)^\top \left(\mathcal{H}\left(\theta_t + yz\left(\theta_{t+1} - \theta_t\right)\right) - \mathcal{H}\left(\theta_t\right)\right)\left(\theta_{t+1} - \theta_t\right) dydz}_{Z}$$

with the first and third equalities due to the fundamental theorem of calculus.

Mark the Hessian eigendecompositions as follows:

$$\mathcal{H}\left(\theta_t + yz\left(\theta_{t+1} - \theta_t\right)\right) = \tilde{V}\tilde{\Lambda}\tilde{V}^\top$$

$$\mathcal{H}\left(\theta_t\right) = V\Lambda V^\top$$

with diagonal $\Lambda = diag\left(\lambda_i\right)_{i=1}^n, \tilde{\Lambda} = diag\left(\tilde{\lambda}_i\right)_{i=1}^n$ and orthogonal (due to the Hermitian nature of Hessian matrices) matrices $V, \tilde{V}$.

$$Z = \int_0^1 \int_0^1 y\left(\theta_{t+1} - \theta_t\right)^\top \left(\tilde{V}\tilde{\Lambda}\tilde{V}^\top - V\Lambda V^\top\right)\left(\theta_{t+1} - \theta_t\right) dydz$$

$$= \int_0^1 \int_0^1 y\left(\theta_{t+1} - \theta_t\right)^\top \left(V\tilde{\Lambda}V^\top - V\Lambda V^\top\right)\left(\theta_{t+1} - \theta_t\right) dydz$$

$$+ \int_0^1 \int_0^1 y\left(\theta_{t+1} - \theta_t\right)^\top \left(\tilde{V}\tilde{\Lambda}\tilde{V}^\top - V\tilde{\Lambda}V^\top\right)\left(\theta_{t+1} - \theta_t\right) dydz$$

Focusing on the first term,

$$= \int_0^1 \int_0^1 y\left(\theta_{t+1} - \theta_t\right)^\top V\left(\tilde{\Lambda} - \Lambda\right)V^\top \left(\theta_{t+1} - \theta_t\right) dydz$$

$$= \int_0^1 \int_0^1 y \sum_{j=1}^n \sum_{i=1}^n \left(\theta_{t+1} - \theta_t\right)^\top v_i \cdot \left(\theta_{t+1} - \theta_t\right)^\top v_j \cdot v_j^\top V\left(\tilde{\Lambda} - \Lambda\right)V^\top v_i dydz$$

$$= \int_0^1 \int_0^1 y \cdot \sum_{i=1}^n \left(\left(\theta_{t+1} - \theta_t\right)^\top v_i\right)^2 \cdot \left(\tilde{\lambda}_i - \lambda_i\right) dydz$$

$$\leq \sum_{i=1}^n L_t^i \cdot \left|\left(\theta_{t+1} - \theta_t\right)^\top v_i\right|^3 \cdot \int_0^1 \int_0^1 y \cdot yz \cdot dydz$$

$$= \sum_{i=1}^n \frac{L_t^i}{6} \cdot \left|\left(\theta_{t+1} - \theta_t\right)^\top v_i\right|^3$$

As for the second term,

$$\int_0^1 \int_0^1 y \left(\theta_{t+1} - \theta_t\right)^\top \left(\tilde{V} \tilde{\Lambda} \tilde{V}^\top - V \tilde{\Lambda} V^\top\right) \left(\theta_{t+1} - \theta_t\right) dy dz$$

$$= \int_0^1 \int_0^1 y \left(\theta_{t+1} - \theta_t\right)^\top \left(\tilde{V} - V\right) \tilde{\Lambda} \left(\tilde{V} + V\right)^\top \left(\theta_{t+1} - \theta_t\right) dy dz$$

$$+ \int_0^1 \int_0^1 y \left(\theta_{t+1} - \theta_t\right)^\top \left(\left(\tilde{V} \tilde{\Lambda} V^\top\right)^\top - \tilde{V} \tilde{\Lambda} V^\top\right) \left(\theta_{t+1} - \theta_t\right) dy dz$$

$$= \int_0^1 \int_0^1 y \left(\theta_{t+1} - \theta_t\right)^\top \left(\tilde{V} - V\right) \tilde{\Lambda} \left(\tilde{V} + V\right)^\top \left(\theta_{t+1} - \theta_t\right) dy dz$$

$$\leq \int_0^1 \int_0^1 y \cdot \left\|\tilde{V} - V\right\|_2 \cdot \left\|\tilde{\Lambda}\right\|_2 \cdot \left\|\tilde{V} + V\right\|_2 \cdot \left\|\theta_{t+1} - \theta_t\right\|_2^2 \cdot dy dz$$

$$\leq \frac{1}{3} \cdot L_R \left\|\tilde{\Lambda}\right\|_2 \cdot \left\|\theta_{t+1} - \theta_t\right\|_2^3$$

$$= \frac{1}{3} \cdot L_R \left\|\tilde{\Lambda}\right\|_2 \cdot \left\|\left(\theta_{t+1} - \theta_t\right) V^\top\right\|_2^3$$

$$\leq \frac{\sqrt{n}}{3} \cdot L_R \left\|\tilde{\Lambda}\right\|_2 \cdot \left\|\left(\theta_{t+1} - \theta_t\right) V^\top\right\|_3^3$$

$$= \frac{\sqrt{n}}{3} \cdot L_R \left\|\tilde{\Lambda}\right\|_2 \cdot \sum_{i=1}^n \left|\left(\theta_{t+1} - \theta_t\right) V^\top \cdot e_i\right|^3$$

$$= \sum_{i=1}^n \frac{\sqrt{n}}{3} \cdot L_R \left\|\tilde{\Lambda}\right\|_2 \cdot \left|\left(\theta_{t+1} - \theta_t\right) v_i\right|^3$$

with

- the first 4 transfers similar to those in the proof of lemma H.4
- the third equality due to orthonormality
- the third inequality due to the $L_p$ norms inequality
- $e_i$ indicating the 1-hot vector with a 1 in the $i$-th entry

Putting it all together (and representing $\theta_{t+1} - \theta_t$ by its coordinate vector over the eigenbasis of $\mathcal{H}(\theta_t)$):

$$f\left(\theta_{t+1}\right) - f\left(\theta_t\right) \leq \sum_{i=1}^n \nabla f\left(\theta_t\right)^\top v_i \cdot \left(\theta_{t+1} - \theta_t\right)^\top v_i + \lambda_i\left(\theta_t\right) \cdot \left(\left(\theta_{t+1} - \theta_t\right)^\top v_i\right)^2$$

$$+ \left(\frac{L_t^i}{6} + \frac{\sqrt{n}}{3} \cdot L_R \cdot L_H\right) \cdot \left|\left(\theta_{t+1} - \theta_t\right) v_i\right|^3$$

$\square$

To understand the relationship between our assumption 2 and the more standard equation 2, we further prove that the combination of assumption 2 and a bounded spectrum assumption will be no weaker than equation 2:

**Lemma H.3.** *Let $A \in \mathbb{R}^{n \times n}, v \in \mathbb{R}^n$. Then $v^\top \left(A^\top - A\right) v = 0$.*

*Proof.*

$$v^\top \left(A^\top - A\right) v = \left(v^\top \left(A^\top - A\right) v\right)^\top = v^\top \left(A - A^\top\right) v = -v^\top \left(A^\top - A\right) v$$

$\square$

**Theorem H.4.** *Let $f : \mathbb{R}^n \to \mathbb{R}$ be a function satisfying assumptions 1 and 2, and assume $\exists_{\lambda_{\sup} \in \mathbb{R}} \forall_{\theta \in \mathbb{R}^n} \forall_{i \in [n]} : \lambda_i(\theta) \leq \lambda_{\sup}$. Then equation 2 is satisfied.*

*Proof.*

$$
\begin{aligned}
\left| p^\top \left( \mathcal{H}(\theta) - \mathcal{H}(\varphi) \right) p \right| &= \left| p^\top \left( V \Lambda V^\top - \tilde{V} \tilde{\Lambda} \tilde{V}^\top \right) p \right| \\
&\leq \left| p^\top \left( V \Lambda V^\top - \tilde{V} \Lambda \tilde{V}^\top \right) p \right| + \left| p^\top \tilde{V} \left( \Lambda - \tilde{\Lambda} \right) \tilde{V}^\top p \right| \\
&= \left| p^\top \left( V - \tilde{V} \right) \Lambda \left( V + \tilde{V} \right)^\top p \right| + \left| p^\top \tilde{V} \left( \Lambda - \tilde{\Lambda} \right) \tilde{V}^\top p \right| \\
&\leq \left\| V - \tilde{V} \right\|_2 \cdot \| \Lambda \|_2 \cdot \left\| V + \tilde{V} \right\|_2 \cdot \| p \|_2^2 + \left\| \tilde{V}^\top p \right\|_2^2 \cdot \left\| \Lambda - \tilde{\Lambda} \right\|_2 \\
&\leq \left( 2 L_R \cdot \sup_{\theta' \in \mathbb{R}^n, i \in \mathbb{R}} \lambda_i(\theta') \right) \cdot \| \theta - \varphi \|_2 + \max_i L^i \cdot \| \theta - \varphi \|_2 \\
&\leq \left( 2 L_H \cdot \lambda_{\sup} + L_H \right) \cdot \| \theta - \varphi \|_2
\end{aligned}
$$

with

- the first inequality due to the triangle inequality

- the second equality due to lemma H.3

- the second inequality due to the Cauchy-Schwartz inequality

- the third inequality due to the triangle inequality, and the fact that all of an orthonormal matrix's eigenvalues equal one of $\{-1, 1\}$

$\square$

## H.3 THEOREM 3.2: WORST CASE-OPTIMAL DESCENT RATE

Before we can prove theorem 3.2, we need to upper bound equation 7.

**Lemma H.5.** *Minmax stepsize bound*

*If $\lambda_i \geq 0$ then*

$$
\Delta \theta_t^{*\top} v_i = \mathcal{O} \left( \sqrt{\left| \nabla f(\theta_t)^\top v_i \right|} \right)
$$

*If $\lambda_i < 0$ then*

$$
\Delta \theta_t^{*\top} v_i = \mathcal{O} \left( |\lambda_i| \right)
$$

*Proof.* For $i$ s.t. $0 \leq \lambda_i \leq \sqrt{L_t^i \cdot \left| \nabla f \left( \theta_t \right)^\top v_i \right|}$ we use corollary H.1.1 with $x = 2L_t^i \cdot \left| \nabla f \left( \theta_t \right)^\top v_i \right|$ to obtain

$$\Delta \theta_t^{*\top} v_i = \frac{\sqrt{\lambda_i^2 + 2L_t^i \cdot \left| \nabla f \left( \theta_t \right)^\top v_i \right|} - \lambda_i}{L_t^i}$$

$$\leq \sqrt{2} \cdot \sqrt{\frac{\left| \nabla f \left( \theta_t \right)^\top v_i \right|}{L_t^i}} + \frac{\frac{\lambda_i^2}{2} \cdot \frac{1}{\sqrt{2L_t^i \cdot \left| \nabla f(\theta_t)^\top v_i \right|}} - \lambda_i}{L_t^i}$$

$$\leq \sqrt{2} \cdot \sqrt{\frac{\left| \nabla f \left( \theta_t \right)^\top v_i \right|}{L_t^i}} + \frac{1}{2\sqrt{2}} \cdot \sqrt{\frac{\left| \nabla f \left( \theta_t \right)^\top v_i \right|}{L_t^i}}$$

$$= \frac{5}{2\sqrt{2}} \cdot \sqrt{\frac{\left| \nabla f \left( \theta_t \right)^\top v_i \right|}{L_t^i}}$$

For $i$ s.t. $\lambda_i > \sqrt{L_t^i \cdot \left| \nabla f \left( \theta_t \right)^\top v_i \right|}$ we use corollary H.1.1 with $x = \lambda_i^2$ to obtain

$$\Delta \theta_t^{*\top} v_i \leq \frac{\left| \nabla f \left( \theta_t \right)^\top v_i \right|}{\lambda_i} \leq \frac{\left| \nabla f \left( \theta_t \right)^\top v_i \right|}{\sqrt{L_t^i \cdot \left| \nabla f \left( \theta_t \right)^\top v_i \right|}} = \sqrt{\frac{\left| \nabla f \left( \theta_t \right)^\top v_i \right|}{L_t^i}}$$

For $i$ s.t. $\lambda_i < 0$, we again use corollary H.1.1 with $x = \lambda_i^2$ to obtain

$$\Delta \theta_t^{*\top} v_i = \frac{2L_t^i \cdot \left| \nabla f \left( \theta_t \right)^\top v_i \right|}{L_t^i \left( \sqrt{\lambda_i^2 + 2L_t^i \cdot \left| \nabla f \left( \theta_t \right)^\top v_i \right|} - |\lambda_i| \right)} \leq \frac{2 |\lambda_i|}{L_t^i} = \mathcal{O} \left( |\lambda_i| \right)$$

$$\square$$

We are now ready to prove theorem 3.2.

**Theorem.** *Worst case-optimal descent rate Let $f$ be a function with Lipschitz-continuous Hessian. After $t$ iterations, algorithm* ELMO *satisfies*

$$f \left( \theta_0 \right) - f \left( \theta_t \right) = \mathcal{O} \left( \log t \right) \tag{17}$$

*Proof.* Cartis et al. (2012a) give $\left| \nabla f \left( \theta_t \right)^\top v_i \right| = \mathcal{O} \left( \frac{1}{t^{\frac{2}{3}}} \right)$ and $\forall_{i:\lambda_i < 0} : |\lambda_i| = \mathcal{O} \left( \frac{1}{\sqrt[3]{t}} \right)$ for the ARC optimization algorithm, of which algorithm ELMO is a special case (the case where ARC perfectly estimates the Hessian Lipschitz parameter).

Making use of lemma H.5 and noting that $m_t^i \left( \Delta \theta_t^{*\top} v_i \right) \leq 0$ by equation 13:

$$\left| m_t^i \left( \Delta \theta_t^{*\top} v_i \right) \right| = \left| \nabla f \left( \theta_t \right)^\top v_i \right| \cdot \Delta \theta_t^{*\top} v_i + \frac{-\lambda_i}{2} \cdot \left( \Delta \theta_t^{*\top} v_i \right)^2 + \frac{L_t^i}{6} \cdot \left( \Delta \theta_t^{*\top} v_i \right)^3$$

For $i$ s.t. $\lambda_i \geq 0$:

$$\leq \left| \nabla f \left( \theta_t \right)^\top v_i \right| \cdot \mathcal{O} \left( \sqrt{\frac{\left| \nabla f \left( \theta_t \right)^\top v_i \right|}{L_t^i}} \right) + \mathcal{O} \left( \sqrt{\frac{\left| \nabla f \left( \theta_t \right)^\top v_i \right|}{L_t^i}} \right)^3$$

$$= \mathcal{O} \left( \left| \nabla f \left( \theta_t \right)^\top v_i \right|^{1.5} \right) = \mathcal{O} \left( \left( \frac{1}{t^{\frac{2}{3}}} \right)^{1.5} \right) = \mathcal{O} \left( \frac{1}{t} \right)$$

For $i$ s.t. $\lambda_i < 0$:

$$\leq \left| \nabla f \left( \theta_t \right)^\top v_i \right| \cdot \mathcal{O} \left( |\lambda_i| \right) + |\lambda_i| \cdot \left( \mathcal{O} \left( |\lambda_i| \right) \right)^2 + \mathcal{O} \left( |\lambda_i| \right)^3 = \mathcal{O} \left( \left| \nabla f \left( \theta_t \right)^\top v_i \right| \cdot |\lambda_i| + |\lambda_i|^3 \right)$$

$$= \mathcal{O} \left( \frac{1}{t^{\frac{2}{3}}} \cdot \frac{1}{\sqrt[3]{t}} + \frac{1}{t} \right) = \mathcal{O} \left( \frac{1}{t} \right)$$

Finally, we have

$$f \left( \theta_0 \right) - f \left( \theta_T \right) = \sum_{t=0}^{T-1} f \left( \theta_t \right) - f \left( \theta_{t+1} \right)$$

$$\leq \sum_{t=0}^{T-1} \left| m_t^i \left( \Delta \theta_t^{*\top} v_i \right) \right|$$

$$\leq \sum_{t=1}^{T-1} \mathcal{O} \left( \frac{1}{t} \right)$$

$$\leq \mathcal{O} \left( \log t + \gamma + \frac{1}{2t} \right)$$

$$= \mathcal{O} \left( \log t \right)$$

with $\gamma \approx 0.57721$ as the Euler-Mascheroni constant and Young (1991) for the last inequality. $\square$

## H.4 THEOREM 3.3

**Theorem.**
$$\left| m_t^i \left( \Delta \theta_t^{*\top} v_i \right) \right| \leq 5 \left| M_t^i \left( \Delta \theta_t^{*\top} v_i \right) \right|$$

*Proof.* Due to equation 13, we have $\frac{m_t^i \left( \Delta \theta_t^{*\top} v_i \right)}{M_t^i \left( \Delta \theta_t^{*\top} v_i \right)} = \left| \frac{m_t^i \left( \Delta \theta_t^{*\top} v_i \right)}{M_t^i \left( \Delta \theta_t^{*\top} v_i \right)} \right|$. Now:

$$\frac{m_t^i \left( \Delta \theta_t^{*\top} v_i \right)}{M_t^i \left( \Delta \theta_t^{*\top} v_i \right)}$$

$$= \frac{\nabla f \left( \theta_t \right)^\top v_i \cdot \Delta \theta_t^{*\top} v_i + \frac{\lambda_i}{2} \left( \Delta \theta_t^{*\top} v_i \right)^2 - \frac{L_t^i}{6} \cdot \left| \Delta \theta_t^{*\top} v_i \right|^3}{\nabla f \left( \theta_t \right)^\top v_i \cdot \Delta \theta_t^{*\top} v_i + \frac{\lambda_i}{2} \left( \Delta \theta_t^{*\top} v_i \right)^2 + \frac{L_t^i}{6} \cdot \left| \Delta \theta_t^{*\top} v_i \right|^3}$$

$$= \frac{- \left| \nabla f \left( \theta_t \right)^\top v_i \right| + \frac{\lambda_i}{2} \frac{\sqrt{\lambda_i^2 + 2 L_t^i \left| \nabla f \left( \theta_t \right)^\top v_i \right|} - \lambda_i}{L_t^i} - \frac{L_t^i}{6} \cdot \left( \frac{\sqrt{\lambda_i^2 + 2 L_t^i \left| \nabla f \left( \theta_t \right)^\top v_i \right|} - \lambda_i}{L_t^i} \right)^2}{- \left| \nabla f \left( \theta_t \right)^\top v_i \right| + \frac{\lambda_i}{2} \frac{\sqrt{\lambda_i^2 + 2 L_t^i \left| \nabla f \left( \theta_t \right)^\top v_i \right|} - \lambda_i}{L_t^i} + \frac{L_t^i}{6} \cdot \left( \frac{\sqrt{\lambda_i^2 + 2 L_t^i \left| \nabla f \left( \theta_t \right)^\top v_i \right|} - \lambda_i}{L_t^i} \right)^2}$$

$$= \frac{5 \left( \lambda_i \sqrt{\lambda_i^2 + 2 L_t^i \left| \nabla f \left( \theta_t \right)^\top v_i \right|} - \lambda_i^2 \right) - 8 L_t^i \left| \nabla f \left( \theta_t \right)^\top v_i \right|}{\left( \lambda_i \sqrt{\lambda_i^2 + 2 L_t^i \left| \nabla f \left( \theta_t \right)^\top v_i \right|} - \lambda_i^2 \right) - 4 L_t^i \left| \nabla f \left( \theta_t \right)^\top v_i \right|}$$

If $\lambda_i = 0$, then

$$\frac{m_t^i \left( \Delta \theta_t^{*\top} v_i \right)}{M_t^i \left( \Delta \theta_t^{*\top} v_i \right)} = 2$$

If $\lambda_i > 0$, then

$$\frac{m_t^i\left(\Delta\theta_t^{*\top} v_i\right)}{M_t^i\left(\Delta\theta_t^{*\top} v_i\right)} = \frac{5\dfrac{\sqrt{1+2\frac{L_t^i|\nabla f(\theta_t)^\top v_i|}{\lambda_i^2}}-1}{\frac{L_t^i|\nabla f(\theta_t)^\top v_i|}{\lambda_i^2}}-8}{\dfrac{\sqrt{1+2\frac{L_t^i|\nabla f(\theta_t)^\top v_i|}{\lambda_i^2}}-1}{\frac{L_t^i|\nabla f(\theta_t)^\top v_i|}{\lambda_i^2}}-4} \le \lim_{x\to\infty}\frac{5\frac{\sqrt{1+2x}-1}{x}-8}{\frac{\sqrt{1+2x}-1}{x}-4} = 2$$

due to the monotonic increasing nature of $\psi_5 : \mathbb{R}^+ \to \mathbb{R}, \psi_5(x) = \frac{5\frac{\sqrt{1+2x}-1}{x}-8}{\frac{\sqrt{1+2x}-1}{x}-4}$.

If $\lambda_i < 0$, then

$$\frac{m_t^i\left(\Delta\theta_t^{*\top} v_i\right)}{M_t^i\left(\Delta\theta_t^{*\top} v_i\right)} = \frac{5\dfrac{\sqrt{1+2\frac{L_t^i|\nabla f(\theta_t)^\top v_i|}{|\lambda_i|^2}}+1}{\frac{L_t^i|\nabla f(\theta_t)^\top v_i|}{|\lambda_i|^2}}+8}{\dfrac{\sqrt{1+2\frac{L_t^i|\nabla f(\theta_t)^\top v_i|}{|\lambda_i|^2}}+1}{\frac{L_t^i|\nabla f(\theta_t)^\top v_i|}{|\lambda_i|^2}}+4} \le \lim_{x\to 0^+}\frac{5\frac{\sqrt{1+2x}+1}{x}+8}{\frac{\sqrt{1+2x}+1}{x}+4} = 5$$

due to the monotonic decreasing nature of $\psi_6 : \mathbb{R}^+ \to \mathbb{R}, \psi_6(x) = \frac{5\frac{\sqrt{1+2x}+1}{x}+8}{\frac{\sqrt{1+2x}+1}{x}+4}$. $\qquad\square$

## H.5    THEOREM 4.2: WORST-CASE DESCENT RATE FOR ARBITRARY OPTIMIZERS

**Theorem.** *Relative Descent*

- 
$$\left|\frac{M_t^i\left(\Delta\theta_t^\top v_i\right) - M_t^i\left(\Delta\theta_t^{*\top} v_i\right)}{M_t^i\left(\Delta\theta_t^{*\top} v_i\right)}\right| = \Theta\left(\left|\Delta\Delta^i\theta_t'\right|^2\right)$$

- 
$$\left|\frac{m_t^i\left(\Delta\theta_t^\top v_i\right) - m_t^i\left(\Delta\theta_t^{*\top} v_i\right)}{m_t^i\left(\Delta\theta_t^{*\top} v_i\right)}\right| = \Theta\left(\left|\Delta\Delta^i\theta_t'\right|\right) \tag{18}$$

*Proof.* For the first part of the lemma,

$$
\left| \frac{M_t^i \left( \Delta \theta_t^\top v_i \right) - M_t^i \left( \Delta \theta_t^{*\top} v_i \right)}{M_t^i \left( \Delta \theta_t^{*\top} v_i \right)} \right|
$$

$$
= \frac{\nabla f \left( \theta_t \right)^\top \cdot v_i \cdot \left( \left( \theta_{t+1} - \theta_t \right)^\top v_i - \Delta \theta_t^{*\top} v_i \right)}{\Delta \theta_t^{*\top} v_i \cdot \nabla f \left( \theta_t \right)^\top v_i + \frac{1}{2} \lambda_i \left( \Delta \theta_t^{*\top} v_i \right)^2 + \frac{L_t^i}{6} \left( \Delta \theta_t^{*\top} v_i \right)^3}
$$

$$
+ \frac{\frac{1}{2} \lambda_i \cdot \left( \left( \left( \theta_{t+1} - \theta_t \right)^\top v_i \right)^2 - \left( \Delta \theta_t^{*\top} v_i \right)^2 \right)}{\Delta \theta_t^{*\top} v_i \cdot \nabla f \left( \theta_t \right)^\top v_i + \frac{1}{2} \lambda_i \left( \Delta \theta_t^{*\top} v_i \right)^2 + \frac{L_t^i}{6} \left( \Delta \theta_t^{*\top} v_i \right)^3}
$$

$$
+ \frac{\frac{L_t^i}{6} \cdot \left( \left( \left( \theta_{t+1} - \theta_t \right)^\top \cdot v_i \right)^3 - \left( \Delta \theta_t^{*\top} v_i \right)^3 \right)}{\Delta \theta_t^{*\top} v_i \cdot \nabla f \left( \theta_t \right)^\top v_i + \frac{1}{2} \lambda_i \left( \Delta \theta_t^{*\top} v_i \right)^2 + \frac{L_t^i}{6} \left( \Delta \theta_t^{*\top} v_i \right)^3}
$$

$$
= \Delta \Delta^i \theta_t' \left( \frac{\frac{L_t^i}{6} \cdot \left( \Delta \Delta^i \theta_t'^2 + 2 \Delta \Delta^i \theta_t' + 3 \right) \cdot \left( \Delta \theta_t^{*\top} v_i \right)^2}{\frac{L_t^i}{6} \left( \Delta \theta_t^{*\top} v_i \right)^2 + \frac{1}{2} \lambda_i \Delta \theta_t^{*\top} v_i - \left| \nabla f \left( \theta_t \right)^\top v_i \right|} \right.
$$

$$
+ \frac{\lambda_i \cdot \left( \frac{1}{2} \Delta \Delta^i \theta_t' + 1 \right) \cdot \Delta \theta_t^{*\top} v_i}{\frac{L_t^i}{6} \left( \Delta \theta_t^{*\top} v_i \right)^2 + \frac{1}{2} \lambda_i \Delta \theta_t^{*\top} v_i - \left| \nabla f \left( \theta_t \right)^\top v_i \right|}
$$

$$
\left. - \frac{\left| \nabla f \left( \theta_t \right)^\top \cdot v_i \right|}{\frac{L_t^i}{6} \left( \Delta \theta_t^{*\top} v_i \right)^2 + \frac{1}{2} \lambda_i \Delta \theta_t^{*\top} v_i - \left| \nabla f \left( \theta_t \right)^\top v_i \right|} \right)
$$

$$
= \Delta \Delta^i \theta_t' \left( \frac{\frac{1}{6} \cdot \left( \Delta \Delta^i \theta_t'^2 + 2 \Delta \Delta^i \theta_t' + 3 \right) \cdot \frac{\left( \sqrt{\lambda_i^2 + 2 L_t^i \cdot \left| \nabla f(\theta_t)^\top v_i \right|} - \lambda_i \right)^2}{L_t^i}}{\frac{1}{6} \frac{\left( \sqrt{\lambda_i^2 + 2 L_t^i \cdot \left| \nabla f(\theta_t)^\top v_i \right|} - \lambda_i \right)^2}{L_t^i} + \frac{1}{2} \lambda_i \frac{\sqrt{\lambda_i^2 + 2 L_t^i \cdot \left| \nabla f(\theta_t)^\top v_i \right|} - \lambda_i}{L_t^i} - \left| \nabla f \left( \theta_t \right)^\top v_i \right|} \right.
$$

$$
+ \frac{\lambda_i \cdot \left( \frac{1}{2} \Delta \Delta^i \theta_t' + 1 \right) \cdot \left( \frac{\sqrt{\lambda_i^2 + 2 L_t^i \cdot \left| \nabla f(\theta_t)^\top v_i \right|} - \lambda_i}{L_t^i} \right)}{\frac{1}{6} \frac{\left( \sqrt{\lambda_i^2 + 2 L_t^i \cdot \left| \nabla f(\theta_t)^\top v_i \right|} - \lambda_i \right)^2}{L_t^i} + \frac{1}{2} \lambda_i \frac{\sqrt{\lambda_i^2 + 2 L_t^i \cdot \left| \nabla f(\theta_t)^\top v_i \right|} - \lambda_i}{L_t^i} - \left| \nabla f \left( \theta_t \right)^\top v_i \right|}
$$

$$
\left. - \frac{\left| \nabla f \left( \theta_t \right)^\top \cdot v_i \right|}{\frac{1}{6} \frac{\left( \sqrt{\lambda_i^2 + 2 L_t^i \cdot \left| \nabla f(\theta_t)^\top v_i \right|} - \lambda_i \right)^2}{L_t^i} + \frac{1}{2} \lambda_i \frac{\sqrt{\lambda_i^2 + 2 L_t^i \cdot \left| \nabla f(\theta_t)^\top v_i \right|} - \lambda_i}{L_t^i} - \left| \nabla f \left( \theta_t \right)^\top v_i \right|} \right)
$$

$$
= \Delta \Delta^i \theta_t' \cdot \left( \frac{\Delta \Delta^i \theta_t' + 2 \Delta \Delta^i \theta_t'^2}{\lambda_i \sqrt{\lambda_i^2 + 2 L_t^i \cdot \left| \nabla f \left( \theta_t \right)^\top v_i \right|} - \lambda_i^2 - 4 L_t^i \left| \nabla f \left( \theta_t \right)^\top v_i \right|} \cdot \lambda_i^2 \right.
$$

$$
+ \frac{2 \left( \Delta \Delta^i \theta_t'^2 + 2 \Delta \Delta^i \theta_t' \right)}{\lambda_i \sqrt{\lambda_i^2 + 2 L_t^i \cdot \left| \nabla f \left( \theta_t \right)^\top v_i \right|} - \lambda_i^2 - 4 L_t^i \left| \nabla f \left( \theta_t \right)^\top v_i \right|} \cdot L_t^i \cdot \left| \nabla f \left( \theta_t \right)^\top v_i \right|
$$

$$
\left. - \frac{\Delta \Delta^i \theta_t' + 2 \Delta \Delta^i \theta_t'^2}{\lambda_i \sqrt{\lambda_i^2 + 2 L_t^i \cdot \left| \nabla f \left( \theta_t \right)^\top v_i \right|} - \lambda_i^2 - 4 L_t^i \left| \nabla f \left( \theta_t \right)^\top v_i \right|} \cdot \lambda_i \sqrt{\lambda_i^2 + 2 L_t^i \cdot \left| \nabla f \left( \theta_t \right)^\top v_i \right|} \right)
$$

If $\lambda_i = 0$:

$$= -\Delta\Delta^i\theta_t'^2 \cdot \left(1 + \frac{1}{2}\Delta\Delta^i\theta_t'^2\right)$$

If $\lambda_i > 0$:

$$= \Delta\Delta^i\theta_t'^2 \cdot \frac{1}{\frac{1}{\sqrt{1+2\frac{L_t^i \cdot |\nabla f(\theta_t)^\top v_i|}{\lambda_i^2}}+1} - 2}$$

$$\cdot \left( \left(\Delta\Delta^i\theta_t' + 2\right) - \left(1 + 2\Delta\Delta^i\theta_t'\right) \cdot \frac{1}{1 + \sqrt{1+2\frac{L_t^i \cdot |\nabla f(\theta_t)^\top v_i|}{\lambda_i^2}}} \right)$$

$$= \Delta\Delta^i\theta_t'^2 \cdot \left( \Delta\Delta^i\theta_t' - 1 - \sqrt{1 + 2\frac{L_t^i \cdot \left|\nabla f(\theta_t)^\top v_i\right|}{\lambda_i^2}} \cdot \left(\Delta\Delta^i\theta_t' + 2\right) \right)$$

$$\cdot \frac{1}{1 + 2\sqrt{1+2\frac{L_t^i \cdot |\nabla f(\theta_t)^\top v_i|}{\lambda_i^2}}}$$

$$= \Delta\Delta^i\theta_t'^3 \cdot \frac{1}{1 + 2\sqrt{1+2\frac{L_t^i \cdot |\nabla f(\theta_t)^\top v_i|}{\lambda_i^2}}}$$

$$- \Delta\Delta^i\theta_t'^2 \cdot \left( 1 + \frac{1}{2}\left(1 - \frac{1}{1 + 2\sqrt{1+2\frac{L_t^i \cdot |\nabla f(\theta_t)^\top v_i|}{\lambda_i^2}}}\right) \right) \cdot \Delta\Delta^i\theta_t'$$

$$= \left( \frac{3}{2}\frac{1}{1 + 2\sqrt{1+2\frac{L_t^i \cdot |\nabla f(\theta_t)^\top v_i|}{\lambda_i^2}}} - \frac{1}{2} \right) \cdot \Delta\Delta^i\theta_t'^3 - \Delta\Delta^i\theta_t'^2$$

Proving that

$$\frac{1}{1 + 2\sqrt{1+2\frac{L_t^i \cdot |\nabla f(\theta_t)^\top v_i|}{\lambda_i^2}}} \in \left(0, \frac{1}{3}\right]$$

would conclude the proof for this case. This is easily proven, by noting that

$$\psi_1 : \mathbb{R}^+ \to \mathbb{R}, \psi_1(x) = \left(1 + 2\sqrt{1+2x}\right)^{-1}$$

is monotonic and satisfies

$$\lim_{x \to 0^+} \psi_1(x) = \frac{1}{3}$$

$$\lim_{x \to \infty} \psi_1(x) = 0$$

If, on the other hand, $\lambda_i < 0$:

$$= -\Delta\Delta^i\theta_t'^2 \cdot \left( \left(1 + 2\Delta\Delta^i\theta_t'\right) - \frac{3}{2}\Delta\Delta^i\theta_t' \cdot \frac{\frac{L_t^i\left|\nabla f(\theta_t)^\top v_i\right|}{|\lambda_i|^2}}{\frac{L_t^i\left|\nabla f(\theta_t)^\top v_i\right|}{|\lambda_i|^2} + \frac{1}{4}\sqrt{1 + 2\frac{L_t^i \cdot \left|\nabla f(\theta_t)^\top v_i\right|}{|\lambda_i|^2}} + \frac{1}{4}} \right)$$

$$= \left( \frac{3}{2} \cdot \frac{\frac{L_t^i\left|\nabla f(\theta_t)^\top v_i\right|}{|\lambda_i|^2}}{\frac{L_t^i\left|\nabla f(\theta_t)^\top v_i\right|}{|\lambda_i|^2} + \frac{1}{4}\sqrt{1 + 2\frac{L_t^i \cdot \left|\nabla f(\theta_t)^\top v_i\right|}{|\lambda_i|^2}} + \frac{1}{4}} - 2 \right)\Delta\Delta^i\theta_t'^3 - \Delta\Delta^i\theta_t'^2$$

Proving that

$$\frac{\frac{L_t^i\left|\nabla f(\theta_t)^\top v_i\right|}{|\lambda_i|^2}}{\frac{L_t^i\left|\nabla f(\theta_t)^\top v_i\right|}{|\lambda_i|^2} + \frac{1}{4}\sqrt{1 + 2\frac{L_t^i \cdot \left|\nabla f(\theta_t)^\top v_i\right|}{|\lambda_i|^2}} + \frac{1}{4}} \in [0, 1)$$

would conclude the proof for this case as well. This is easily proven, by noting that

$$\psi_2 : \mathbb{R}^+ \to \mathbb{R}, \psi_2\left(x\right) = \frac{x}{x + \frac{1}{4}\sqrt{1 + 2x} + \frac{1}{4}}$$

is monotonic and satisfies

$$\lim_{x \to 0^+} \psi_2\left(x\right) = 0$$

$$\lim_{x \to \infty} \psi_2\left(x\right) = 1$$

For the second part of the lemma,

$$
\left| \frac{m_t^i \left( \Delta\theta_t^\top v_i \right) - m_t^i \left( \Delta\theta_t^{*\top} v_i \right)}{m_t^i \left( \Delta\theta_t^{*\top} v_i \right)} \right|
$$

$$
= \frac{\nabla f\left(\theta_t\right)^\top \cdot v_i \cdot \left( \left(\theta_{t+1} - \theta_t\right)^\top v_i - \Delta\theta_t^{*\top} v_i \right)}{\Delta\theta_t^{*\top} v_i \cdot \nabla f\left(\theta_t\right)^\top v_i + \frac{1}{2}\lambda_i \left(\Delta\theta_t^{*\top} v_i\right)^2 - \frac{L_t^i}{6}\left(\Delta\theta_t^{*\top} v_i\right)^3}
$$

$$
+ \frac{\frac{1}{2}\lambda_i \cdot \left( \left(\left(\theta_{t+1} - \theta_t\right)^\top v_i\right)^2 - \left(\Delta\theta_t^{*\top} v_i\right)^2 \right)}{\Delta\theta_t^{*\top} v_i \cdot \nabla f\left(\theta_t\right)^\top v_i + \frac{1}{2}\lambda_i \left(\Delta\theta_t^{*\top} v_i\right)^2 - \frac{L_t^i}{6}\left(\Delta\theta_t^{*\top} v_i\right)^3}
$$

$$
- \frac{\frac{L_t^i}{6} \cdot \left( \left(\left(\theta_{t+1} - \theta_t\right)^\top \cdot v_i\right)^3 - \left(\Delta\theta_t^{*\top} v_i\right)^3 \right)}{\Delta\theta_t^{*\top} v_i \cdot \nabla f\left(\theta_t\right)^\top v_i + \frac{1}{2}\lambda_i \left(\Delta\theta_t^{*\top} v_i\right)^2 - \frac{L_t^i}{6}\left(\Delta\theta_t^{*\top} v_i\right)^3}
$$

$$
= \Delta\Delta^i\theta_t' \left( \frac{\frac{-L_t^i}{6} \cdot \left(\Delta\Delta^i\theta_t'^2 + 2\Delta\Delta^i\theta_t' + 3\right)}{-\frac{L_t^i}{6}\left(\Delta\theta_t^{*\top} v_i\right)^2 + \frac{1}{2}\lambda_i \Delta\theta_t^{*\top} v_i - \left|\nabla f\left(\theta_t\right)^\top v_i\right|} \cdot \left(\Delta\theta_t^{*\top} v_i\right)^2 \right.
$$

$$
+ \frac{\lambda_i \cdot \left(\frac{1}{2}\Delta\Delta^i\theta_t' + 1\right)}{-\frac{L_t^i}{6}\left(\Delta\theta_t^{*\top} v_i\right)^2 + \frac{1}{2}\lambda_i \Delta\theta_t^{*\top} v_i - \left|\nabla f\left(\theta_t\right)^\top v_i\right|} \cdot \Delta\theta_t^{*\top} v_i
$$

$$
\left. - \frac{\left|\nabla f\left(\theta_t\right)^\top \cdot v_i\right|}{-\frac{L_t^i}{6}\left(\Delta\theta_t^{*\top} v_i\right)^2 + \frac{1}{2}\lambda_i \Delta\theta_t^{*\top} v_i - \left|\nabla f\left(\theta_t\right)^\top v_i\right|} \right)
$$

$$
= \Delta\Delta^i\theta_t' \left( \frac{-\frac{1}{6} \cdot \left(\Delta\Delta^i\theta_t'^2 + 2\Delta\Delta^i\theta_t' + 3\right) \cdot \frac{\left(\sqrt{\lambda_i^2 + 2L_t^i \cdot \left|\nabla f(\theta_t)^\top v_i\right|} - \lambda_i\right)^2}{L_t^i}}{-\frac{1}{6}\frac{\left(\sqrt{\lambda_i^2 + 2L_t^i \cdot \left|\nabla f(\theta_t)^\top v_i\right|} - \lambda_i\right)^2}{L_t^i} + \frac{1}{2}\lambda_i \frac{\sqrt{\lambda_i^2 + 2L_t^i \cdot \left|\nabla f(\theta_t)^\top v_i\right|} - \lambda_i}{L_t^i} - \left|\nabla f\left(\theta_t\right)^\top v_i\right|} \right.
$$

$$
+ \frac{\lambda_i \cdot \left(\frac{1}{2}\Delta\Delta^i\theta_t' + 1\right) \cdot \left( \frac{\sqrt{\lambda_i^2 + 2L_t^i \cdot \left|\nabla f(\theta_t)^\top v_i\right|} - \lambda_i}{L_t^i} \right)}{-\frac{1}{6}\frac{\left(\sqrt{\lambda_i^2 + 2L_t^i \cdot \left|\nabla f(\theta_t)^\top v_i\right|} - \lambda_i\right)^2}{L_t^i} + \frac{1}{2}\lambda_i \frac{\sqrt{\lambda_i^2 + 2L_t^i \cdot \left|\nabla f(\theta_t)^\top v_i\right|} - \lambda_i}{L_t^i} - \left|\nabla f\left(\theta_t\right)^\top v_i\right|}
$$

$$
\left. - \frac{\left|\nabla f\left(\theta_t\right)^\top \cdot v_i\right|}{-\frac{1}{6}\frac{\left(\sqrt{\lambda_i^2 + 2L_t^i \cdot \left|\nabla f(\theta_t)^\top v_i\right|} - \lambda_i\right)^2}{L_t^i} + \frac{1}{2}\lambda_i \frac{\sqrt{\lambda_i^2 + 2L_t^i \cdot \left|\nabla f(\theta_t)^\top v_i\right|} - \lambda_i}{L_t^i} - \left|\nabla f\left(\theta_t\right)^\top v_i\right|} \right)
$$

$$= \Delta\Delta^i\theta'_t \left( \frac{12\lambda_i \cdot \sqrt{\lambda_i^2 + 2L_t^i \cdot \left|\nabla f\left(\theta_t\right)^\top v_i\right|} - 12\lambda_i^2 - 12L_t^i \left|\nabla f\left(\theta_t\right)^\top \cdot v_i\right|}{5\lambda_i\sqrt{\lambda_i^2 + 2L_t^i \cdot \left|\nabla f\left(\theta_t\right)^\top v_i\right|} - 5\lambda_i^2 - 8L_t^i \cdot \left|\nabla f\left(\theta_t\right)^\top v_i\right|} \right.$$

$$+ \frac{7\lambda_i \cdot \left(\sqrt{\lambda_i^2 + 2L_t^i \cdot \left|\nabla f\left(\theta_t\right)^\top v_i\right|} - \lambda_i\right) - 4L_t^i \cdot \left|\nabla f\left(\theta_t\right)^\top v_i\right|}{5\lambda_i\sqrt{\lambda_i^2 + 2L_t^i \cdot \left|\nabla f\left(\theta_t\right)^\top v_i\right|} - 5\lambda_i^2 - 8L_t^i \cdot \left|\nabla f\left(\theta_t\right)^\top v_i\right|} \cdot \Delta\Delta^i\theta'_t$$

$$\left. + \frac{2\lambda_i\sqrt{\lambda_i^2 + 2L_t^i \cdot \left|\nabla f\left(\theta_t\right)^\top v_i\right|} - 2\lambda_i^2 - 2L_t^i \cdot \left|\nabla f\left(\theta_t\right)^\top v_i\right|}{5\lambda_i\sqrt{\lambda_i^2 + 2L_t^i \cdot \left|\nabla f\left(\theta_t\right)^\top v_i\right|} - 5\lambda_i^2 - 8L_t^i \cdot \left|\nabla f\left(\theta_t\right)^\top v_i\right|} \cdot \Delta\Delta^i\theta'^2_t \right)$$

Noting the common structure of each of the coefficients of $\Delta\Delta^i\theta'^1_t, \Delta\Delta^i\theta'^2_t, \Delta\Delta^i\theta'^3_t$, we prove the following to bound all three via appropriate settings of $a, b \in \{2, 4, 7, 12\}$:

If $\lambda_i > 0$:

$$\lim_{\frac{L_t^i \cdot \left|\nabla f(\theta_t)^\top v_i\right|}{\lambda_i^2} \to \mathcal{L}} \frac{a\lambda_i \cdot \left(\sqrt{\lambda_i^2 + 2L_t^i \cdot \left|\nabla f\left(\theta_t\right)^\top v_i\right|} - \lambda_i\right) - bL_t^i \cdot \left|\nabla f\left(\theta_t\right)^\top v_i\right|}{5\lambda_i\sqrt{\lambda_i^2 + 2L_t^i \cdot \left|\nabla f\left(\theta_t\right)^\top v_i\right|} - 5\lambda_i^2 - 8L_t^i \cdot \left|\nabla f\left(\theta_t\right)^\top v_i\right|}$$

$$= \lim_{\frac{L_t^i \cdot \left|\nabla f(\theta_t)^\top v_i\right|}{\lambda_i^2} \to \mathcal{L}} \frac{a\frac{2}{\sqrt{1+2\frac{L_t^i \cdot \left|\nabla f(\theta_t)^\top v_i\right|}{\lambda_i^2}}+1} - b}{5\frac{2}{\sqrt{1+2\frac{L_t^i \cdot \left|\nabla f(\theta_t)^\top v_i\right|}{\lambda_i^2}}+1} - 8}$$

$$= \begin{cases} \frac{b-a}{3} & \mathcal{L} = 0^+ \\ \frac{b}{8} & \mathcal{L} = \infty \end{cases}$$

If $\lambda_i \leq 0$:

$$\lim_{\frac{L_t^i \cdot \left|\nabla f(\theta_t)^\top v_i\right|}{\lambda_i^2} \to \mathcal{L}} \frac{a\left|\lambda_i\right| \cdot \left(\sqrt{\left|\lambda_i\right|^2 + 2L_t^i \cdot \left|\nabla f\left(\theta_t\right)^\top v_i\right|} + \left|\lambda_i\right|\right) + bL_t^i \cdot \left|\nabla f\left(\theta_t\right)^\top v_i\right|}{5\left|\lambda_i\right|\left(\sqrt{\left|\lambda_i\right|^2 + 2L_t^i \cdot \left|\nabla f\left(\theta_t\right)^\top v_i\right|} + \left|\lambda_i\right|\right) + 8L_t^i \cdot \left|\nabla f\left(\theta_t\right)^\top v_i\right|}$$

$$= \lim_{\frac{L_t^i \cdot \left|\nabla f(\theta_t)^\top v_i\right|}{\lambda_i^2} \to \mathcal{L}} \frac{a\left(\sqrt{1 + 2\frac{L_t^i \cdot \left|\nabla f(\theta_t)^\top v_i\right|}{\left|\lambda_i\right|^2}} + 1\right) + b\frac{L_t^i \cdot \left|\nabla f(\theta_t)^\top v_i\right|}{\left|\lambda_i\right|^2}}{5\left(\sqrt{1 + 2\frac{L_t^i \cdot \left|\nabla f(\theta_t)^\top v_i\right|}{\left|\lambda_i\right|^2}} + 1\right) + 8\frac{L_t^i \cdot \left|\nabla f(\theta_t)^\top v_i\right|}{\left|\lambda_i\right|^2}}$$

$$= \begin{cases} \frac{a}{5} & \mathcal{L} = 0^+ \\ \frac{b}{8} & \mathcal{L} = \infty \end{cases}$$

Analogously to the first case, and due to the monotonic natures (for all $a, b \in \mathbb{R}$) of

$$\psi_3 : \mathbb{R}^+ \to \mathbb{R}, \psi_3\left(x\right) = \frac{a\frac{2}{\sqrt{1+2x}+1} - b}{5\frac{2}{\sqrt{1+2x}+1} - 8}$$

and

$$\psi_4 : \mathbb{R}^+ \to \mathbb{R}, \psi_4\left(x\right) = \frac{a\left(\sqrt{1+2x} + 1\right) + bx}{5\left(\sqrt{1+2x} + 1\right) + 8x}$$

the term in the parentheses is bounded, thus we may conclude our proof of the lemma. $\square$

**Remark.** *Note that when $\lambda_i > 0$, $\frac{L_t^i \cdot |\nabla f(\theta_t)^\top v_i|}{\lambda_i^2} \to 0^+$, the coefficients of ${\Delta\Delta^i \theta_t'}^3, {\Delta\Delta^i \theta_t'}^1$ shrink to 0 (since $a = b$ for those cases), so that $\left| \frac{m_t^i(\Delta\theta_t^\top v_i) - m_t^i(\Delta\theta_t^{*\top} v_i)}{m_t^i(\Delta\theta_t^{*\top} v_i)} \right| = \Theta\left( {\Delta\Delta^i \theta_t'}^2 \right)$*

We are now ready to prove theorem 4.2.

**Theorem.** *Worst-case descent rate for arbitrary optimizers*

*Let $f : \mathbb{R}^n \to \mathbb{R}$ a twice-differentiable function satisfying assumptions 1 and 2, and let $\Delta\theta_t$ satisfy $M_t^i\left(\Delta\theta_t^\top v_i\right) \le 0$. Then*

- 

$$\left| \frac{M_t^i\left(\Delta\theta_t^\top v_i\right)}{M_t^i\left(\Delta\theta_t^{*\top} v_i\right)} \right| = \Theta\left( 1 + \left|\Delta\Delta^i \theta_t'\right|^2 \right)$$

- 

$$\left| \frac{m_t^i\left(\Delta\theta_t^\top v_i\right)}{m_t^i\left(\Delta\theta_t^{*\top} v_i\right)} \right| = \Theta\left( 1 + \left|\Delta\Delta^i \theta_t'\right|^p \right) \tag{19}$$

*with $p = \begin{cases} 2 & \lambda_i > 0 \wedge \frac{|\nabla f(\theta_t)^\top v_i|}{\lambda_i^2} = 0 \\ 1 & else \end{cases}$.*

*Proof.* Proof is immediate from lemma H.5, because we have

$$\frac{M_t^i\left(\Delta\theta_t^\top v_i\right)}{M_t^i\left(\Delta\theta_t^{*\top} v_i\right)} = 1 + \frac{M_t^i\left(\Delta\theta_t^\top v_i\right) - M_t^i\left(\Delta\theta_t^{*\top} v_i\right)}{M_t^i\left(\Delta\theta_t^{*\top} v_i\right)}$$

and similarly for $m_t^i\left(\Delta\theta_t^\top v_i\right)$. $\qquad\square$

# I   LIMITATIONS AND FUTURE WORK

One interesting direction for future research is in putting the estimated Lipschitz parameters to work throughout the optimization process to increase the descent rate in hopes of matching and even surpassing ARC's strong performance (Xu et al., 2017). Although the code attached to this paper is capable of estimating these parameters, it does so too slowly to be practically useful in computing all of an algorithm's steps, under most settings. We suggest future work could improve this algorithm's computational complexity.

A limitation of our Newton's method performance predictor is the additional computational burden of computing the Lipschitz parameters. We provide code for doing so in the attached code on Github, but we recommend performing these computations sparingly, since the Lipschitz parameters are approximately locally stable anyway.

A second limitation of our work is its inability to provide any indication of the number of iterations left to achieve convergence. We see this as an acceptable limitation however, since in practice a model is only required to achieve a certain level of performance on the data decided ahead of time, without regard to how much further it could be optimized. As noted in the introduction, performance is measured by the loss function, so our descent rate bound satisfies this practical requirement.

A final limitation of our bound is its reliance on $\Delta\Delta^i \theta_t$ as a measure of algorithm optimality which is a function of $\Delta\theta_t^{*\top} v_i$, despite the fact that most optimizers do not compute that during training. This bound is therefore primarily of theoretical interest, as illustrated by its motivation of the very practical metric discussed in section 6

