# OpenReview forum: "Universal Concavity-Aware Descent Rate for Optimizers"
_ICLR.cc/2025/Conference — Submitted to ICLR 2025_

### Official Review · Reviewer_xG9h · 2024-10-23

**Soundness:** 2
**Presentation:** 2
**Contribution:** 1
**Rating:** 3
**Confidence:** 4

**Summary:**

The paper presents a Hessian based optimization algorithm which optimizes the step size coordinate-wise depending on the magnitude of the Lipschitz constant of the objective function along the eigen space coordinates.

**Strengths:**

Unfortunately, the paper is not convincing enough to demonstrate that the main idea could be useful, so I cannot see any strengths.

**Weaknesses:**

Besides that Hessian-based optimization techniques are often already too resource intense for large scale optimization, the paper suggest to perform a quite costly extra eigen-decomposition in every optimization step and assumes the existence of a "Lipschitz oracle" which can accurately estimate the Lipschitz constant of the objective function along every coordinate in the eigen space. Besides very simple objective functions I cannot see how this can be done efficiently. The paper provides Algorithm 2 which seems to be a full optimization algorithm (replacement of ELMO?, not clear) which uses an unexplained "BASE_OPT" subroutine and the "ASSESS_LIPSCHITZ" subroutine which depends on the unexplained (optimal?) point $\theta_t^*$.

I also cannot see that the Lipschitz constant of Assumption 2 is actually a magnitude smaller than the one defined in equation (2). The paper promises to discuss this by saying "we will demonstrate that the Lipschitz parameters relevant to these subspaces are often order of magnitude smaller than the others", but provides no reference and I could not locate any such discussion.

The paper seems to provide a general optimization algorithm, but the introduction heavily motivates its application for hyper parameter optimization. In that domain the objective function is quite expensive to evaluate, so I especially cannot see how one can possess a Lipschitz oracle there. Furthermore, the assumptions of requiring an objective function of Lipschitz continuous Hessian (even coordinstewise) is usually violated in the hyper parameter optimization domain, and even in the experimental examples studied by the paper for neural network training with RELU activation functions in Appendix F.

There are a lot of confusing and wrong statements. For example in Lemma 3.1 "Exists $(L_t^i) : i=1,...,n$" does not make sense to me, the reformulation of $M_t^i(x)$ at the bottom of page 5 is certainly wrong (and seems to be completely out of context anyway) because whenever $x \cdot v_i(\theta_t)$ is nonzero that maximum is infinite, the $\arg \min$ in (7) returns a vector but it should be equal to scalars on its left and right sides, $\theta_{t+1} \leftarrow ...$ does not seem to depend directly on $\theta_t$ at all (is that really so?), Theorem 3.2 hides and did not discuss the dependence on $L_H$ which would have been one of the major points of the paper, and Theorem 3.3 is claimed to demonstrate that the upper bound of Theorem 3.2 matches the lower bound, but I cannot see how this result is related to any performance lower bound and it is neither discussed there.

The writing style is neither very clear including statements like "step's distance from ELMO's step relative to ELMO's step" in Notation 8, "only on subspaces with significantly convex subspaces" on page 8, and "Logarithmic scale" for Figure 1 which does not seem to be logarithmic.

**Questions:**

Could you tell me where is this promised demonstration for "we will demonstrate that the Lipschitz parameters relevant to these subspaces are often order of magnitude smaller than the others" in the paper?

---

> ### Author Response · Authors · 2024-11-14
>
> Thank you for your in-depth critique. We'll address the points you brought up below:
>
> Computational complexity:
> Although it is true that the ELMO algorithm is computationally complex, it is not intended to be a practical optimization algorithm, but instead an "ideal" algorithm (in the minmax sense) for other algorithms to compare themselves to. Previous works provide various different methods for computing the step to take at each iteration, often producing positive definite preconditioner matrices which usually attempt to justify themselves through some connection to the inverse Hessian (e.g. Saddle-Free Newton, FOSI, natural gradient methods including Adagrad and KFAC), but insufficient work has been done on defining what the best preconditioner would be in the nonconvex setting; we propose the ELMO algorithm to close that gap, and provide tools (based on Theorem 4.2) for demonstrating how close arbitrary optimization algorithms are to this "ideal algorithm". As such, we don't recommend performing optimization directly with ELMO, but instead - perform a small number of iterations periodically throughout the optimization process or only at the beginning of the optimization process (which may be executed with an arbitrary optimization algorithm) to determine how well the current algorithm is doing relative to what ELMO could have achieved, and adjust the hyperparameters (or even use a different algorithm altogether) accordingly. Alternatively, one may forego ELMO iterations altogether and instead use a heuristic we noted in section 5.1 which notes that convex Lipschitz constants are primarily affected by the modelling task, and not model design/loss function/data.
>
> Furthermore, although it may seem impractical in its current form, Lanczos iterations have commonly been used in the practical optimization literature for Hessian eigendecomposition (usually run periodically with large intervals in between), and the ARC optimization algorithm is notable for estimating the Lipschitz parameters adaptively; future work may cause direct ELMO approximations to become computationally feasible. We hope that providing a theoretically optimal algorithm will help future researchers develop new and improved second-order optimization algorithms, which is becoming a pressing need as progress slows on increasing the speed of first-order method iterations.
>
> Algorithm 2 describes our method for computing Lipschitz parameters (acting as the Lipschitz oracle you mentioned), it is an adaptation of the well known ARC algorithm (Nesterov & Polyak, 2006, "Cubic regularization of newton method and its global performance") which also computes Lipschitz parameters. You are correct about the BASE_OPT subroutine, we will clarify that in the final draft; we are referring to gradient descent/Adam (experiment dependent), following Sivan et al. (2024)'s notation. The ASSESS_LIPSCHITZ subroutine is not dependent on the optimal point $\theta^*$, that is instead the ELMO step as defined in equation (7); we will improve that notation in the final draft.
>
> Lipschitz distribution (and Question): The Lipschitz parameters from Assumption 2 are compared experimentally against the one from equation 2 in section 5 and in appendix F, and compared theoretically in appendix H.2. Our Hessian Lipschitz continuity assumption is not stronger than that prevalent in previous works (and confirmed in practice in prior work), see the discussion and comparison in appendices E and H.2.
>
> Confusing formulations: We very much appreciate any feedback you can give us for further clarification before the final draft.
> - The notation in Lemma 3.1 claims that "there exist $n$ positive real scalars marked $L_t^i$ satisfying...".
> - The maximum is not infinite due to the Lipschitz continuity assumption upper bounding the maximal function value for a given value of $x$ (see Lemma 3.1 which bounds the function value).
> - The left and right sides of equation (7) are scalars for a particular $t$ and $i$ (the left side is a dot product of vectors, and the right side is a ratio of scalars). The summation symbol in the middle is a typo however, and will be corrected in the final draft.
> - You are correct, the update of $\theta_{t+1}$ in algorithm ELMO is a typo and should begin with $\theta_t + \dots$
> - Theorem 3.2 is not a central part of the paper, and its dependence on $L_H$ is irrelevant to its descent rate
> - ELMO lower bound: You are correct, we will clarify that further in the final draft. The idea was that we proved in Lemma 3.1 that $f(θ_0​)−f(θ_t)$ is bounded between $m$ and $M$, ELMO maximizes $\|M\|$, and $\|m\|$ is within a constant multiplicative factor of $\|M\|$, so that Theorem 3.2's upper bound on $\|M\|$ provides a good estimation of $f(θ_0​)−f(θ_t)$ (to within a constant multiplicative factor).
> - "Significantly convex subspaces" refers to eigenspaces of the Hessian whose corresponding eigenvalue is very large.
> - Figure 1 is logarithmic (look at the $y$-axis scale)

---

> > ### Comment · Reviewer_xG9h · 2024-11-25
> > **Thank you for the answer, I will keep my original score.**
> >
> > Thank you for the answer, but unfortunately I will keep my original score.
> >
> > Some reflections to your comments:
> > - There is no Lipschitz constraint in the formulation of the maximum at the bottom of page 5, so I think you should adjust your notation to reflect that.
> > - If ELMO is not a practical algorithm as you say, then your paper should be "strong" on the theoretical side. And then I believe Theorem 3.2 and the role of $L_H$ are important. I still believe it would be important in a paper like this to see at least an example which shows that $L_H$ of Assumption 2 is indeed significantly smaller than the uniform smoothness constant on the Hessian.

---

> ### Comment · Area_Chair_pDsn · 2024-11-24
>
> Dear Reviewer xG9h,
>
> The author discussion phase will be ending soon. The authors have provided detailed responses. Could you please reply to the authors with whether they have addressed your concern and whether you will keep or modify your assessment on this submission?
>
> Thanks.
>
> Area Chair

---

### Official Review · Reviewer_5Upg · 2024-11-01

**Soundness:** 2
**Presentation:** 2
**Contribution:** 1
**Rating:** 3
**Confidence:** 4

**Summary:**

The paper introduces a new second-order optimization method that achieves the bound f(\theta_0) - f(\theta_t) = O(log t).  The method involves the eigendecomposition of the Hessian matrix at the current iterate, computes the Lipschitz constants of the Hessian in the eigenvector directions, and uses these to determine an optimal step that minimizes a third-order Taylor polynomial model. Experimental results indicate that second-order optimizers outperform first-order optimizers when the Lipschitz constants of the convex eigenspace of the Hessian are relatively small.

**Strengths:**

The premise of the paper is correct, namely, that it is difficult to compare algorithms across a broad spectrum of problem classes.

**Weaknesses:**

1) The paper notes in Appendix C, Line 1034, that “Cauchy’s method (Traub, 1982) is nearly identical to ELMO.” However, if this is the case, Cauchy’s method (Traub, 1982) should be reviewed in the main body rather than the Appendix, with a clear emphasis on the differences between it and the ELMO method.
2) Although the introduction (Line 84) claims to “bound the rate at which the model’s performance increases (as measured by the loss function),” Theorem 3.2 provides only a negative lower bound for f(θt​)−f(θ0​), which seems weak. It’s unclear how Theorem 3.3 contributes to addressing this negative lower bound.
3) The performance of the ELMO algorithm is not demonstrated in the experiments.
4) The difficulty described in the introduction (Lines 50–65) does not seem to serve as a motivation for the ELMO method. Specifically, ELMO's design and analysis do not appear to address the challenge of selecting the best optimizer among a variety of options, each with its own assumptions and convergence rates.
5) Overall, the connection between the introduction (motivation), the ELMO method design, the experiments, and the conclusion is weak.

**Questions:**

1) For Equation (7): Why is the positive root taken? When compute the derivative of |\Delta {theta^*_t}^T v_i|^3 in (3), why is it assumed that  \Delta {theta^*_t}^T v_i is positive?
2) In line 377-379: Are there typos where “concave” should be “convex”?

---

> ### Author Response · Authors · 2024-11-14
>
> Thank you for your in-depth critique. We'll address the points you brought up below:
>
> Cauchy's algorithm: Perhaps we overstated the similarity to Cauchy's algorithm; we'll correct that in the final draft. While the implementation in 1 dimension is similar, the algorithms differ both in their goals and domains. Cauchy's algorithm's goal is finding the zeros of the gradient in 1-dimensional nonconvex functions, while ELMO's goal is finding the minimum of a nonconvex $n$-dimensional function (via eigendecomposition into $n$ 1-dimensional functions). Note that the zeros of the gradient are not necessarily minima - they may be maxima or shoulder/saddle points (in $n$-dimensional loss functions).
>
> ELMO lower bound: You are correct, we will clarify that further in the final draft.  The idea was that we proved in Lemma 3.1 that $f(θ_0​)−f(θ_t)$ is bounded between $m$ and $M$, ELMO maximizes $\|M\|$, and $\|m\|$ is within a constant multiplicative factor of $\|M\|$ (and by design, $\|m\| \ge \|M\|$), so that Theorem 3.2's upper bound on $\|M\|$ provides a good estimation of $f(θ_0​)−f(θ_t)$ (to within a constant multiplicative factor).
>
> ELMO performance: We stress that ELMO is not intended to be a practical optimization algorithm, but instead - an "ideal" algorithm (in the minmax sense) for other nonconvex optimization algorithms to compare themselves to; therefore, a demonstration of ELMO being used in practice is out-of-scope and beyond our computational budget.
>
> Optimizer selection: While there are many factors involved in selecting an optimizer as noted in the introduction, our work limits itself to helping practitioners select hyperparameters and an optimizer for their training scenario.  A tradeoff often exists between the number of iterations an optimizer requires to achieve a desired performance level and their per-iteration computational burden, however it is not clear from previous work how much computational burden is justified by a given algorithm's effectiveness.  As noted above, we approach that challenge by developing an effectiveness metric based on an "ideal" algorithm (in the minmax sense) for other optimizers to compare themselves to - the closer they are to ELMO (as measured by Notation 7), the more computational burden they are worth.  We will clarify this point further in the final draft.
>
> Questions:
> 1. We wish to minimize $\|M_t^i\|$ by choosing a step size in each eigenspace $span\left(v_i\right)$; this is given by the projection of an arbitrary step $\Delta \theta$ onto this eigenspace, which we can write as $\Delta \theta^\top v_i$.  Finding this step size requires us to compute the root of the derivative of $\|M\|$ w.r.t. $\|\Delta \theta\|$.  We take the positive root since it is a local minimum of $\|M_t^i\|$ (which is to be expected, since when the gradient is negative in this eigenspace as assured by equation (1) line 128, a negative step implies a step in the direction of the gradient, causing the loss function to *rise* in this subspace instead of drop).  This may alternatively be seen by noting that taking the negative root instead of the positive root will cause the equivalent preconditioner given in Lemma 4.1 not to be postiive semi-definite, as required by all quasi-Newton algorithms.
>
> 2. You are correct, those are typos.  We will fix them in the final draft.

---

> ### Comment · Area_Chair_pDsn · 2024-11-24
>
> Dear Reviewer 5Upg,
>
> The author discussion phase will be ending soon. The authors have provided detailed responses. Could you please reply to the authors with whether they have addressed your concern and whether you will keep or modify your assessment on this submission?
>
> Thanks.
>
> Area Chair

---

> > ### Comment · Reviewer_5Upg · 2024-11-25
> > **Thank you**
> >
> > Thanks to the author(s) for their detailed responses.  My assessment remains the same.

---

### Official Review · Reviewer_nich · 2024-11-02

**Soundness:** 3
**Presentation:** 3
**Contribution:** 2
**Rating:** 5
**Confidence:** 2

**Summary:**

This paper addresses the challenges of parameter optimization for machine learning models, particularly in non-convex settings. The authors aim to simplify the comparison of convergence rates among different optimizers by proposing a unified descent rate bound with broad applicability. This work introduces the Eigenspace-Lipschitz Minmax Optimizer (ELMO), a minmax-optimal algorithm that effectively accounts for concave subspaces where traditional convergence bounds may fail.

The ELMO algorithm utilizes a third-order Taylor polynomial to model loss and demonstrates a worst-case descent rate of $O(\log t)$. The authors also analyze the descent rates of quasi-Newton algorithms in relation to ELMO, emphasizing the importance of considering the distribution of Lipschitz parameters in this context.

**Strengths:**

This paper introduces the Eigenspace-Lipschitz Minmax Optimizer (ELMO), which utilizes a third-order Taylor polynomial to model loss functions in machine learning. This approach extends traditional optimization methods by addressing the complexities of non-convex landscapes. Additionally, the authors' focus on the distribution of Lipschitz parameters offers a potentially novel perspective that has not been widely explored in the existing literature.

**Weaknesses:**

I am not familiar with this research topic, so I may underestimate the authors' contributions. However, I appreciate their discussion on when to use first-order versus second-order algorithms, as well as the impact of the distribution of Lipschitz parameters on algorithm performance.

1. The topic studied by the authors is quite broad, making it challenging to establish a unified framework.
2. While the authors present a framework with some theoretical guarantees, it lacks practical applicability.
3. If the algorithm proposed by the authors is only slightly different from Cauchy's method (Traub, 1982), could they clarify the specific contexts in which each of the two algorithms is best applied?

**Questions:**

1. Line 23 and elsewhere: What is the meaning of "tight" or "tightness"?
2. Line 123: Should it be "from j=1 to j=n"?
3. Line 156, Notation 5. Is it guaranteed that the algorithm will converge? Should we introduce any assumptions to ensure convergence?
4. Line 226, Assumption 2: Does this assumption imply that we need to continuously perform eigenvalue decomposition during each iteration of the algorithm?
5. Line 274, equation (7): Why is there a summation symbol present?
6. Line 290, in Algorithm1: When updating $\theta_{t+1}$, should we include $\theta_t$ on the right-hand side?
7. Line 324, Notation 7 and Line 355 Notation 8. These marks may cause confusion.

---

> ### Author Response · Authors · 2024-11-14
>
> Thank you for your in-depth critique.  We'll address the points you brought up below:
>
> Practicality of ELMO: Although it is true that the ELMO algorithm increases the computational complexity due to its requiring eigendecomposition of the Hessian and Hessian-Lipschitz parameter estimation, it is not intended to be a practical optimization algorithm, but instead - an "ideal" algorithm (in the minmax sense) for other algorithms to compare themselves to. Previous works provide various different methods for computing the step to take at each iteration, often producing positive definite preconditioner matrices which usually attempt to justify themselves through some connection to the inverse Hessian (e.g. Saddle-Free Newton, FOSI, natural gradient methods including Adagrad and KFAC), but insufficient work has been done on defining what the best preconditioner would be in the nonconvex setting; we propose the ELMO algorithm to close that gap, and provide tools (based on Theorem 4.2) for demonstrating how close arbitrary optimization algorithms are to this "ideal algorithm". As such, we don't recommend performing optimization directly with ELMO, but instead - perform a small number of iterations periodically throughout the optimization process or only at the beginning of the optimization process (which may be executed with an arbitrary optimization algorithm) to determine how well the current algorithm is doing relative to what ELMO could have achieved, and adjust the hyperparameters (or even use a different algorithm altogether) accordingly. Alternatively, one may forego ELMO iterations altogether and instead use a heuristic we noted in section 5.1 which notes that convex Lipschitz constants are primarily affected by the modelling task, and not model design/loss function/data. Furthermore, although it may seem impractical in its current form, Lanczos iterations have commonly been used in the practical optimization literature for Hessian eigendecomposition, and the ARC optimization algorithm is notable for estimating the Lipschitz parameters adaptively; future work may cause direct ELMO approximations to become computationally feasible.  We hope that providing a theoretically optimal algorithm will help future researchers develop new and improved second-order optimization algorithms, which is becoming a pressing need as progress slows on increasing the speed of first-order method iterations.
>
> Cauchy's algorithm:
> Perhaps we overstated the similarity to Cauchy's algorithm; we'll correct that in the final draft.  While the implementation in 1 dimension is similar, the algorithms differ both in their goals and domains.  Cauchy's algorithm's goal is finding the zeros of the gradient in 1-dimensional nonconvex functions, while ELMO's goal is finding the minimum of a nonconvex $n$-dimensional function (via eigendecomposition into $n$ 1-dimensional functions).  Note that the zeros of the gradient are not necessarily minima - they may be maxima or shoulder/saddle points (in $n$-dimensional loss functions).
>
> Questions:
> 1. We use "tightness" to quantify how well a convergence rate bound describes the rate at which optimization algorithms reach minima of the loss functions they are applied to.  While technically correct, an upper bound may not provide users with a useful indication of the number of training iterations that will be necessary to achieve the required minimality of the loss function, when the upper bound differs significantly from the true number of iterations that will be necessary.
>
> 2. You are correct, we will fix that in the final draft.
>
> 3. Wang et al., 2017, "Stochastic quasi-newton methods for nonconvex stochastic optimization" provide easily applicable conditions for convergence, however we simply assume that the algorithms a user uses are convergent.  We note that this assumption will not be used in our work in any way besides discussion of prior works.
>
> 4. It does not imply that, however you will need to perform eigendecomposition at each iteration to use algorithm ELMO (which we do not recommend using at each iteration!).  Previous works (e.g. Sivan et al. 2024, "FOSI: Hybrid first and second order optimization") address this computational burden however, by noting that in practice, it need not be performed at every iteration.
>
> 5,6. You are correct, those are typos, we will fix them in the final draft.
>
> 7. You are correct, we will change those to more easily distinguishable notations in the final draft.

---

> > ### Comment · Reviewer_nich · 2024-11-25
> >
> > Thank you for your rebuttal. I am unsure whether ELMO has the potential to contribute to the development of new and improved second-order optimization algorithms. I will keep my score.

---

> ### Comment · Area_Chair_pDsn · 2024-11-24
>
> Dear Reviewer  nich,
>
> The author discussion phase will be ending soon. The authors have provided detailed responses. Could you please reply to the authors with whether they have addressed your concern and whether you will keep or modify your assessment on this submission?
>
> Thanks.
>
> Area Chair

---

### Official Review · Reviewer_Y8Ge · 2024-11-04

**Soundness:** 2
**Presentation:** 2
**Contribution:** 2
**Rating:** 5
**Confidence:** 4

**Summary:**

In this paper the authors discuss a preconditioned gradient method for various convex and non-convex optimization problems.  The main idea is to construct cubic polynomials (similar to cubic regularized Newton method) in d different directions, determined by the eigendirections of the Hessian at the current point. Since eigendirections are orthogonal by construction, this approach enables the authors to derive direction-specific step-sizes. Such a decomposition is made possible by leveraging a Lipschitz assumption of the Hessian's eigenvalues and eigenvectors.

**Strengths:**

The key strength of this paper lies in its novel decomposition of the optimization problem into d separate cubic polynomials along eigendirections, which allows for direction-specific step size optimization. This approach is particularly useful as it transforms what would typically be a d-dimensional cubic optimization problem into d one-dimensional problems that admit closed-form solutions for direction-specific step sizes. By exploiting the orthogonality of eigendirections and the Lipschitz properties of the Hessian's spectral decomposition, the authors provide a computationally tractable solution to adaptive step size selection.

**Weaknesses:**

Theoretical Novelty and Assumptions:

1. While the paper presents an interesting approach, its theoretical foundations are largely built upon existing work in cubic regularized Newton methods and the Lipschitz assumption of eigenvalues and eigenvectors, making the contribution incremental.  The global Lipschitz assumption on eigenvalues and eigenvectors seems overly restrictive, particularly may be problematic in regions where eigenvalues are small, potentially limiting the method's practical applicability.

Implementation and Practicality:

2. It would be nice to have a self-contained description of how to implement the algorithm in practice, particularly regarding the estimation of Lipschitz parameters for eigenvalues. While Section F.1 references a method, it lacks sufficient detail for reproduction; this section refers to some other methods which are not mentioned in the paper.

3. The convergence analysis does not adequately address how the rate might be affected when using practical approximations of the Lipschitz parameters.

4. The computational overhead of the proposed method is significant, and the paper doesn't convincingly demonstrate practical advantages over simpler, widely-used preconditioned gradient methods like Adam or RMSprop that are more straightforward to implement.

In summary, while the paper presents an interesting theoretical framework, its strong assumptions and computational complexity, coupled with unclear practical benefits over existing methods, limit its potential impact on practical applications in machine learning problems.

**Questions:**

Please look at the weakness section.

---

> ### Author Response · Authors · 2024-11-13
>
> Thank you for your in-depth critique.  We'll address the points you brought up below:
>
> Novelty:
> While previous works assumed a single global scalar Lipschitz parameter for the entirety of all Hessians, our contribution is in making a more fine-grained assumption - different Lipschitz parameters for each eigenspace at each iteration.  This is significant since as demonstrated in our experiments (section 5.1 and appendix F), these Lipschitz parameters vary widely, such that selecting a single Lipschitz parameter for all of them would lead to an overly conservative (larger than necessary) Lipschitz parameter for many of the subspaces (in particular the convex subspaces, in which much of the optimization process focuses in many optimizers!  See Gur-Ari et al. 2018, "Gradient descent happens in a tiny subspace").  This overly conservative Lipschitz parameter leads to inefficient steps of the ELMO optimizer by causing them to be smaller than necessary, see appendix D.2.  As such, our assumptions are not stronger than those prevalent in previous works (and confirmed in practice in prior work), see the discussion and comparison in appendices E and H.2.
>
> Estimation algorithm for Lipschitz parameters:
> We provide an adaption of ARC (discovered independently by Andreas Griewank, 1981, "The modification of newton’s method for unconstrained optimization by bounding cubic terms" and Nesterov & Polyak, 2006, "Cubic regularization of newton method and its global performance") in appendix F.1, as cited in the paper.  A discussion of details on hyperparameters selection may be found in those works and our hyperparameters may be found in appendix F.
>
> Effect of practical approximation of Lipschitz parameters on convergence:
> While appendix D discusses the effect of inaccurate estimations of the Lipschitz parameters on the ELMO algorithm and previous works (e.g. Nesterov & Polyak, 2006, "Cubic regularization of newton method and its global performance") analyze the effect of practical Lipschitz estimation on ARC's convergence rate (and their analysis is applicable to ELMO as well), we stress that ELMO is not intended to be an optimization algorithm to be used in practice, but instead should be seen as an "ideal" optimizer (in the minmax sense) to compare arbitrary optimizers to (for an effectiveness analysis of those algorithms), making use of the tools developed in this paper (based on Theorem 4.2).  As such, an analysis of the effect of practical estimations of the Lipschitz parameter on its convergence rate do not contribute to this work (since we do not recommend doing that anyway).
>
> Computational overhead:
> As noted in the previous paragraph, although it is true that the ELMO algorithm increases the computational complexity due to its requiring eigendecomposition of the Hessian and Hessian-Lipschitz parameter estimation, it is not intended to be a practical optimization algorithm, but instead - an "ideal" algorithm (in the minmax sense) for other algorithms to compare themselves to. Previous works provide various different methods for computing the step to take at each iteration, often producing positive definite preconditioner matrices which usually attempt to justify themselves through some connection to the inverse Hessian (e.g. Saddle-Free Newton, FOSI, natural gradient methods including Adagrad and KFAC), but insufficient work has been done on defining what the best preconditioner would be in the nonconvex setting; we propose the ELMO algorithm to close that gap, and provide tools (based on Theorem 4.2) for demonstrating how close arbitrary optimization algorithms are to this "ideal algorithm". As such, we don't recommend performing optimization directly with ELMO, but instead - perform a small number of iterations periodically throughout the optimization process or only at the beginning of the optimization process (which may be executed with an arbitrary optimization algorithm) to determine how well the current algorithm is doing relative to what ELMO could have achieved, and adjust the hyperparameters (or even use a different algorithm altogether) accordingly. Alternatively, one may forego ELMO iterations altogether and instead use a heuristic we noted in section 5.1 which notes that convex Lipschitz constants are primarily affected by the modelling task, and not model design/loss function/data. Furthermore, although it may seem impractical in its current form, Lanczos iterations have commonly been used in the practical optimization literature for Hessian eigendecomposition, and the ARC optimization algorithm is notable for estimating the Lipschitz parameters adaptively; future work may cause direct ELMO approximations to become computationally feasible.  We hope that providing a theoretically optimal algorithm will help future researchers develop new and improved second-order optimization algorithms, which is becoming a pressing need as progress slows on increasing the speed of first-order method iterations.

---

> > ### Comment · Area_Chair_pDsn · 2024-11-24
> >
> > Dear Reviewer  Y8Ge,
> >
> > The author discussion phase will be ending soon. The authors have provided detailed responses. Could you please reply to the authors with whether they have addressed your concern and whether you will keep or modify your assessment on this submission?
> >
> > Thanks.
> >
> > Area Chair

---

### Official Review · Reviewer_HzqY · 2024-11-06

**Soundness:** 2
**Presentation:** 2
**Contribution:** 1
**Rating:** 3
**Confidence:** 3

**Summary:**

This submission introduces a novel concavity-aware optimization framework called ELMO, which adapts the descent rate based on the concavity in different subspaces of the loss function, represented by eigenvalues of the Hessian $\lambda_i$. By leveraging eigenspace-specific Lipschitz constants, $L_i$, the optimizer adjusts its step size dynamically across both convex and concave subspaces, improving descent efficiency. Through theoretical analysis and preliminary experiments, ELMO is shown to provide a versatile and efficient optimization approach with potential advantages over traditional quasi-Newton type methods.

**Strengths:**

- By focusing on concave subspaces, the work explores a new theoretical direction that could inspire new optimizer designs that perform better on complex, non-convex objectives.

- I checked portions of the appendix proofs, and they are generally correct. Their preliminary numerical experiments validate that the analyzed bounds are meaningful.

**Weaknesses:**

- While the paper introduces the ELMO algorithm as a theoretically optimal method, there’s a glaring omission of computational complexity analysis in practical scenarios. Specifically, computing the Hessian’s eigendecomposition and Lipschitz constants is computationally intensive, which limits scalability.

- The paper’s theoretical sections hint at the advantages over other optimizers but stop short of providing robust empirical comparisons. Most notably, results from modern optimizers like L-BFGS or RMSprop are missing. The paper’s single-minded focus on first-order methods as baselines (SGD, etc.) is insufficient, given that it positions ELMO as a quasi-Newton optimizer.

- The authors emphasize concave subspaces within non-convex optimization but seem to overlook other complexities typical in high-dimensional loss landscapes, such as well-known flat regions and saddle points. The argument that concave subspaces alone are sufficient for achieving descent is unconvincing and feels disconnected from empirical evidence on the non-convexity challenges in neural networks.


### small issues:


- the eigendecomposition in notation 2 is $(v_j,\lambda_j)_{j=0}^n$ (or $j=1$)

- Notation 4 should be placed before equation 1

- page 4, what's the difference of your $o(\cdot)$ and $\mathcal{O}(\cdot)$

- assumption 2 and equation 2 should be $L_H\in \mathbb{R}^+$. meanwhile I don't really get your purpose w.r.t. condition $\bar{L}^i\geq 1/L_H$, what's the intuition of this assumption?

- the appearance of $\Delta\Delta$ starting at Notation 7 is somehow weird, why don't you use something like $\Delta^2$ by convention in the difference-equation-related analysis?

- Since $v_i$ and $−v_i$ are both equally viable, if we assume $\forall_{i \in[n]}: \nabla f\left(\theta_t\right)^{\top} v_i \geq 0$ rather than your assumption, we can simplify your proofs in appendix H.3 ~ H.5 by removing the unnecessary representations. Is that? or I missed something

- Appendix A, "generallhy" instead of "generally"

- The proof contains numerous repetitive expressions. The authors may consider using general notations to replace them, enhancing the math readability. Alternatively, including a high-level idea (or proof sketch) in your main body would improve the clarity.

**Questions:**

- The meta-optimizer concept mentioned at the end of the paper is interesting but lacks detail. Can you envision this as an adaptive framework for switching between first- and second-order methods during training? What metrics or criteria would govern these switches, and what evidence suggests that the meta-optimizer approach would be computationally feasible?

- The paper emphasizes descent in concave subspaces. However, most real-world optimization problems (especially neural networks, you claimed) exhibit a mix of convexity, concavity, and flat regions. Could you clarify how ELMO handles such mixed regions? Why was this concavity-only focus chosen, and what empirical evidence supports its sufficiency?

---

> ### Author Response · Authors · 2024-11-13
>
> Thank you for your in-depth critique.  We'll address the point you brought up below:
>
> Practicality of ELMO:
> Although it is true that the ELMO algorithm increases the computational complexity due to its requiring eigendecomposition of the Hessian and Hessian-Lipschitz parameter estimation, it is not intended to be a practical optimization algorithm, but instead - an "ideal" algorithm (in the minmax sense) for other algorithms to compare themselves to.  Previous works provide various different methods for computing the step to take at each iteration, often producing positive definite preconditioner matrices which usually attempt to justify themselves through some connection to the inverse Hessian (e.g. Saddle-Free Newton, FOSI, natural gradient methods including Adagrad and KFAC), but insufficient work has been done on defining what the best preconditioner would be in the nonconvex setting; we propose the ELMO algorithm to close that gap, and provide tools (based on Theorem 4.2) for demonstrating how close arbitrary optimization algorithms are to this "ideal algorithm".  As such, we don't recommend performing optimization directly with ELMO, but instead - perform a small number of iterations periodically throughout the optimization process or only at the beginning of the optimization process (which may be executed with an arbitrary optimization algorithm) to determine how well the current algorithm is doing relative to what ELMO could have achieved, and adjust the hyperparameters (or even use a different algorithm altogether) accordingly.  Alternatively, one may forego ELMO iterations altogether and instead use a heuristic we noted in section 5.1 which notes that convex Lipschitz constants are primarily affected by the modelling task, and not model design/loss function/data.  Furthermore, although it may seem impractical in its current form, Lanczos iterations have commonly been used in the practical optimization literature for Hessian eigendecomposition, and the ARC optimization algorithm is notable for estimating the Lipschitz parameters adaptively; future work may cause direct ELMO approximations to become computationally feasible.  We hope that providing a theoretically optimal algorithm will help future researchers develop new and improved second-order optimization algorithms, which is becoming a pressing need as progress slows on increasing the speed of first-order method iterations.
>
> Lack of experiments with modern optimizers:
> The purpose of the experiments was to demonstrate the tool we developed which helps one choose between first- and second- order algorithms; we felt that this was best achieved with a "pure" first order method (SGD) and a "pure" second-order method (FOSI).  Modern optimizers like RMSProp and LBFGS attempt to get a very crude approximation of the Hessian (see appendix A) at low cost, but while this is practically useful, we see it as a sort of "middle ground" between first- and second- order methods; as such, it does not contribute to the claim we were providing evidence for (the claim that second-order methods are not as effective when the convex Lipschitz parameter is large thus are not worth their additional computational burden).  Furthermore, there is much literature demonstrating the poor generalization capabilities of these methods (e.g. see some of the references from section 1).
>
> Emphasis on concavity:
> We do not claim that concave spaces alone are sufficient for achieving descent, in fact our methods apply equally well to convex and concave subspaces of the parameter space; we merely wished to emphasize the importance of these spaces due to the little attention they've received in past works.  Also, one of the significant contributions of our work is its emphasis on a decomposition of the loss function into n 1-dimensional functions; these must be locally convex or concave (or both, in the case of flat/linear regions).  As such, saddle points are decomposed into dimensions in which they are convex and dimensions in which they are concave, and ELMO can handle each in turn optimally.
>
> Small issues:
> We agree with your corrections, thank you for pointing them out.  We will correct them before the final draft.  The intuition behind L_H is simply to choose L_H large enough to satisfy the conditions.
>
> Questions:
> 1. You correctly understood the idea of the meta-optimizer we propose.  We expect to use the metric from notation 8 and to set thresholds for switching between methods based on a tradeoff between their computational cost and their expected effectiveness (based on the metric from Notation 8).
> 2. As mentioned before, we fully support optimizing in all subspaces, convex and linear included, and our analysis supports that.  We simply wished to emphasize the concave subspaces of the loss function as well, since they've received insufficient attention in past works, in particular when it comes to theoretical justifications for the steps taken in concave subspaces.

---

> ### Comment · Area_Chair_pDsn · 2024-11-24
>
> Dear Reviewer  HzqY,
>
> The author discussion phase will be ending soon. The authors have provided detailed responses. Could you please reply to the authors with whether they have addressed your concern and whether you will keep or modify your assessment of this submission?
>
> Thanks.
>
> Area Chair

---

> ### Comment · Reviewer_HzqY · 2024-11-25
> **Ack.**
>
> Thanks for writing the rebuttal, which unfortunately didn't change my opinion.

---

### Meta-Review · Area_Chair_pDsn · 2024-12-10

**Metareview:**

Multiple authors have expressed a concern on the computational efficiency of ELMO. I understand that the authors do not suggest using ELMO to solve an optimization problem, but instead use it as an ideal method to help the selection of optimization algorithm. However, for problem with a high dimension, performing an eigenvalue decomposition is still intractable so evening performing ELMO periodically or at the beginning is impossible. If so, no information can be provided by ELMO.

I suggest the authors state the value of ELMO more clearly and earlier in the paper, indicating that ELMO is not intended to be an efficient solver but a tool to select algorithm.  It will be also good to test ELMO on a low-dimensional instance where the eigenvalue decomposition can be done in each iteration at a low cost.

**Additional Comments On Reviewer Discussion:**

I gave a lower weight on the review given by Reviewer nich because he/she has a lower confidence score and is not an expert in the area.

All reviewers have concerns on the computational efficiency of the ELMO method in this paper. In addition, Reviewer HzqY pointed out that considering concave subspaces is not enough and saddle points must be considered also. Reviewer Y8Ge had a concern that the convergence analysis does not show how the rate is affected when using practical approximations of the Lipschitz parameters. Reviewer 5Upg indicated that the ELMO algorithm should be included in the experiments (maybe with a low-dimensional example who eigenvalues can be computed efficiently). He or she also was concerned that ELMO does not help selecting the best optimizer. Reviewer xG9h pointed out some confusing and wrong statements

---

### Decision · Program_Chairs · 2025-01-22

Reject